# The atomistic details of the ice recrystallisation inhibition activity of PVA

Fabienne Bachtiger[1,2], Thomas R. Congdon[1], Christopher Stubbs[1], Matthew I. Gibson [1,3] &
Gabriele C. Sosso [1,2✉]

Understanding the ice recrystallisation inhibition (IRI) activity of antifreeze biomimetics is crucial to the development of the next generation of cryoprotectants. In this work, we bring together molecular dynamics simulations and quantitative experimental measurements to unravel the microscopic origins of the IRI activity of poly(vinyl)alcohol (PVA)—the most potent of biomimetic IRI agents. Contrary to the emerging consensus, we find that PVA does not require a "lattice matching" to ice in order to display IRI activity: instead, it is the effective volume of PVA and its contact area with the ice surface which dictates its IRI strength. We also find that entropic contributions may play a role in the ice-PVA interaction and we demonstrate that small block co-polymers (up to now thought to be IRI-inactive) might display significant IRI potential. This work clarifies the atomistic details of the IRI activity of PVA and provides novel guidelines for the rational design of cryoprotectants.

[1] Department of Chemistry, University of Warwick, Coventry, UK. [2] Centre for Scientific Computing, University of Warwick, Coventry, UK. [3] Warwick Medical School, University of Warwick, Coventry, UK. ✉email: g.sosso@warwick.ac.uk

**B**eing able to understand and control ice recrystallisation (IR) is of fundamental importance for a wide range of applications[1,2], not least in the prevention of cold-induced damage of cryopreserved biological materials. IR is a form of Ostwald ripening[3], where larger ice crystals are formed at the expense of smaller ones and typically occurs during the thawing process of a frozen substance. The damage arises because the formation of the larger ice crystals causes osmotic stress and can rupture and pierce cells, resulting in cell death[4]. IR damage therefore poses a threat to the viable recovery of cryobiologically stored material, leading to the waste of precious resources (e.g. cells and tissue samples) used in regenerative medicine[5–7]. IR damage can be mitigated by the use of cryoprotectants, but their usage suffers from extensive work protocols and potential cytotoxicity[5,8–10]. Consequently, a substantial body of work[5,6,10–17] has been devoted to finding alternative cryoprotectants that are biocompatible, relatively straightforward to produce and which display exceptional ice recrystallisation inhibition (IRI) activity at comparatively low concentrations.

The main obstacle to the rational design of new IRI-active compounds is our patchy understanding of their structure-function relation: in fact, even in the case of prototypical IRI agents such as the synthetic polymer poly(vinyl)alcohol (PVA), there is much speculation about the molecular-level interactions between PVA and ice[7,11,12,14,18–21]. PVA has been found to be capable of protecting proteins during freezing, by preventing irreversible protein aggregation on growing ice crystals, as well as enabling solvent-free cryopreservation of bacteria[11,12]. Similarly, PVA has been shown to mitigate IR during cryopreservation and enables some increase in post-thaw recovery for mammalian cells[7]. Thus, understanding the underlying mechanism by which PVA confers IRI activity is key for the development, refining and optimisation of the next-generation cryoprotectants that can be fine-tuned for a diverse range of applications (e.g. regenerative medicine, food storage and crop security[1]).

Recent molecular dynamics simulations[22] have suggested that the IRI activity of PVA might be due to the "distance matching" between the hydroxyl groups of the polymer and water molecules at the ice surface, in which PVA would require to adopt a linear/extended conformation to achieve such a distance match. In a similar vein, computational work by Weng et al.[23] indicated that PVA did not have to bind to ice in a linear conformation via a geometric lattice matching per se: rather, the study showed a stereoscopic match of hydroxyls to water molecules in the ice lattice which then facilitated binding and IRI activity. The IRI properties of PVA seem to be closely linked to its degree of polymerisation (DP), i.e., the number of monomeric units in a polymer chain, with shorter chain PVA (DP = 10) showing a significant reduction in activity compared to PVA with DP = 19[20]. In fact, PVA is thought to be significantly IRI active only for DP values greater or equal than this critical threshold of DP = 19. Recent simulations[24] have rationalised this in terms of a size dependence of the ice-binding efficiency of PVA, showing that short-chain polymers are less likely to bind to ice and that the polymer-ice hydrogen bond life-times for small polymers (DP < 10) is rather short compared to the time taken for ice to grow. Thus, while the microscopic motivations at the heart of the IRI activity of PVA are still hotly debated, it would seem there is a growing consensus that a minimum number of hydroxyl groups is required for PVA to bind ice long enough to stop the growth of an advancing ice front.

In this work, we use molecular dynamic (MD) simulations to investigate the IRI activity of PVA by systematically probing the impact of conformation, number of functional groups and degree of polymerisation (see Fig. 1 for computational setup along with Methods section for further details). In contrast to a number of previous studies[22,24,25], we show that PVA can bind ice in any conformation and that lattice matching is not a prerequisite for ice inhibition. We also show that short-chain PVA$_{10}$ has reduced IRI activity not because it cannot bind ice sufficiently well, but

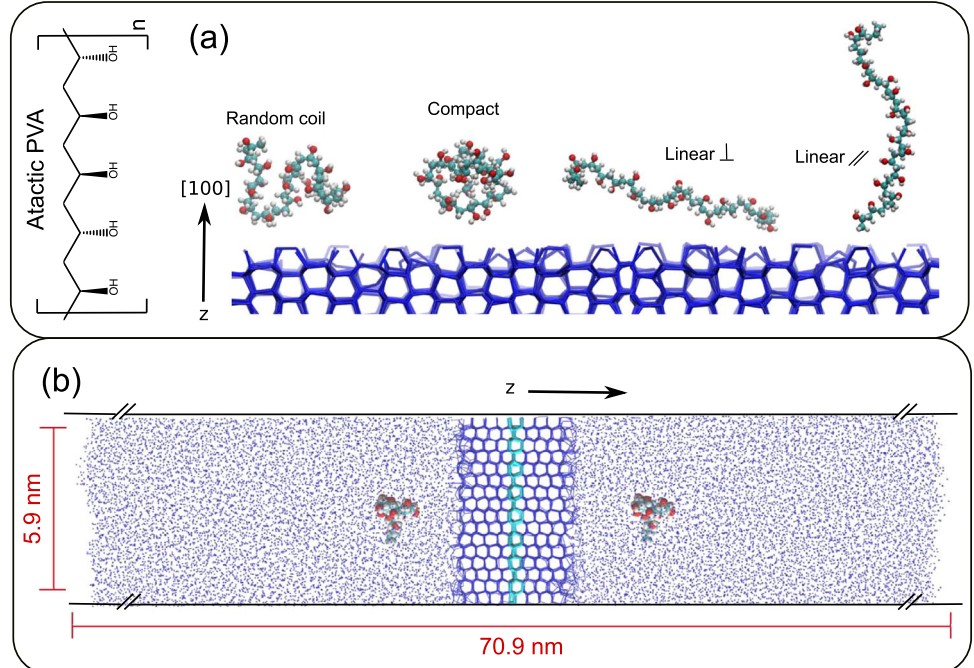

**Fig. 1 Computational set-up. a** Molecular structure of PVA along with representative configurations of the simulated polymers. Random coil refers to the polymer simulations that do not have their radius of gyration ($R_g$) restrained, compact and linear refer to simulations in which the $R_g$ has been restrained to a defined value (see Methods section). For the linear polymers, their orientation with respect to the direction (z) of the growing ice front is also restrained, either perpendicular or parallel to the z-axis. **b** Representative snapshot of the computational geometry. The restrained seeded ice layer is highlighted in cyan. The ice front presenting the primary prismatic face to the water phase grows in the ±z direction.

rather because it becomes overgrown by the ice front and that sustained IRI activity is dependent on the volume occupied by a polymer at the ice interface. We also show that both enthalpic and entropic contributions may play a part in PVA-ice interactions. Lastly, we explore the role small chain block copolymers such as PVA-b-PVAm (polyvinylalcohol/polyvinylamine) could have in designing IRI active polymers with DP as small as 10, which has been previously thought of as unattainable.

## Results

**Lattice matching is not required for PVA to bind to ice**. In order to understand the impact of the PVA conformation on its ice-binding activity, we performed well-tempered metadynamics simulations (see Methods section) using the radius of gyration $R_g$ as the collective variable to bias: the lower the value of $R_g$, the more compact the PVA chain is. As a result, we were able to explore the conformational free energy landscape of PVA (see Fig. 2); we find that the polymer adopts a wide range of different conformations, from very compact ($R_g = 0.5$ nm) to extended ($R_g = 0.95$ nm). The free energy barriers separating the minima are of the same order of thermal fluctuations at room temperature ($k_BT \sim 25$ meV). In addition, the most favourable conformation ($R_g = 0.74$ nm) corresponds to a random coil, which is consistent with what we observe in our unbiased simulations of $PVA_{20}$ at 265 K.

The role of PVA conformation has been a central argument surrounding the mechanistic details of its IRI activity. In particular, previous experimental[26] and theoretical works[22,24,25] have suggested that a linear/extended conformation is crucial for the polymer to adopt in order to bind ice and display IRI activity. The rational for this argument is based on a "lattice matching" of the hydrogen bonding between the hydroxyl functional groups of PVA to the water molecules of the ice crystal. Recent computational work[22] on PVA has suggested a specific 2:1 binding pattern: two adjacent hydroxyls groups bind to ice, followed by an

unbound group; the distance between the two binding hydroxyls is comparable to the nearest oxygen–oxygen distance within the primary prismatic ice surface. Other recent simulations[23] have suggested that the binding between PVA and ice is based on the stereoscopic matching of the hydroxyl groups to the underlying ice lattice. Our metadynamics simulations suggest however that given the very flexible nature of PVA, the probability that this polymer binds to an advancing ice front by adopting a linear/extended conformation is vanishingly small. Furthermore, the lattice matching argument assumes the growing ice front to be a smooth surface: while relatively smooth ice surfaces can develop via a spiral growth regime based on screw dislocations, this scenario is characteristic of much stronger supercooling than those considered here – and indeed is not thought to be relevant in cryopreservation conditions, see e.g. refs. [27–29]. In contrast, when dealing with ice-water interfaces characterised by some degree of roughness (as considered in this work), the probability of PVA binding thanks to specific structural patterns consistent with the ice lattice is even further reduced.

To provide concrete evidence of the role of PVA conformation on its IRI activity, we have performed MD simulations of $PVA_{20}$ where we have either let the polymer free to adopt any conformation or we have restrained it by applying a bias to its radius of gyration so as to enforce a specific orientation of the polymer with respect to the ice surface. We discuss in the Methods section the computational setup as well as the details of the order parameters we have used to identify ice-like molecules. These simulations start with PVA being in solution which is in contact with a seeded ice crystal (see Fig. 1). The simulation is then run over a 200 ns period in which the ice front grows and IRI activity is monitored.

Figure 3 show representative snapshots of these simulations where: (a) an extended PVA chain is restrained so as to be perpendicular with respect to the growing ice front; (b) a similarly extended conformation is restrained so as to be parallel with respect to the growing ice front and; (c) a compact, globular conformation is enforced: it is evident that PVA can bind to ice in any conformation, albeit in the case of conformation (b) the ice phase grows largely unhindered by the presence of the bound polymer. Once bound, the polymers do not detach within the timescale of our simulations (see Supplementary Fig. 3 for further details): while this result obviously holds at this particular supercooling only (8 K), it is indicative that at mild supercooling the binding of PVA might not be reversible – in agreement with the work found in Naullage et al.[22] We note, however, that the binding affinity of the polymer might become relevant in conditions where the kinetics of ice formation is slow enough for PVA to bind reversibly.

These results hold for MD simulations of unrestrained $PVA_{20}$ (i.e. no bias on $R_g$) as well, as illustrated in Supplementary Fig. 4. Thus, we can conclude that indeed the lattice matching argument does not play a significant role in determining the IRI activity of PVA. Previous computational work[22] put forward a 2:1 binding pattern for PVA on ice which we do not observe in our simulations. This contradiction is likely due to the fact that the investigation reported in Naullage et al.[22]: (i) assumed that PVA behaves like a linear molecule when biding to ice, which appears not to be the case at the supercooling investigated here; (ii) utilised the mW[30] and a united-atom force field[31] to model water molecules and PVA, respectively. The coarse-grained nature of these force fields is unlikely to capture the complexity of hydrogen bonding at heterogeneous interfaces such as the PVA-ice one. We use a fully atomistic force field instead, as discussed in the Methods section; (iii) the distance between the two binding hydroxyls characterising this 2:1 pattern was thought to be comparable to the nearest oxygen–oxygen distance

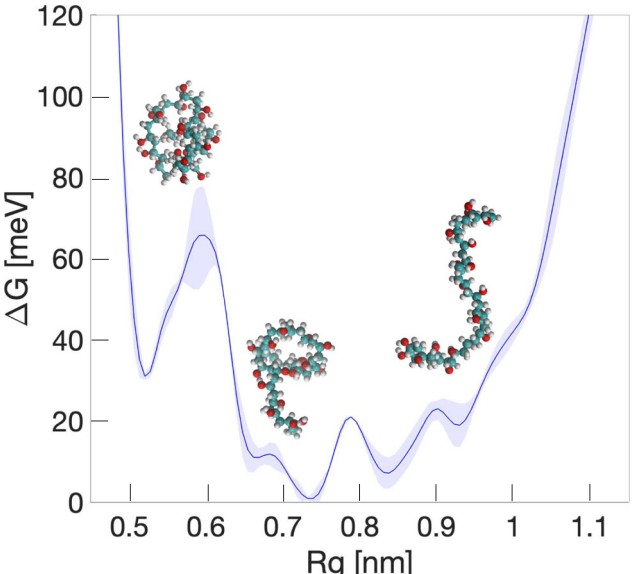

**Fig. 2 Conformational landscape of $PVA_{20}$ at 300 K.** Gibbs free energy ($\Delta G$) profile as a function of the radius of gyration $R_g$, obtained by means of metadynamics simulations. $PVA_{20}$ is a very flexible polymer spanning a substantial range of different conformations, all of them separated by energy barriers of the same order of thermal fluctuations at room temperature ($k_BT \sim 25$ meV). The shaded region represents the standard error associated with our estimate of $\Delta G$.

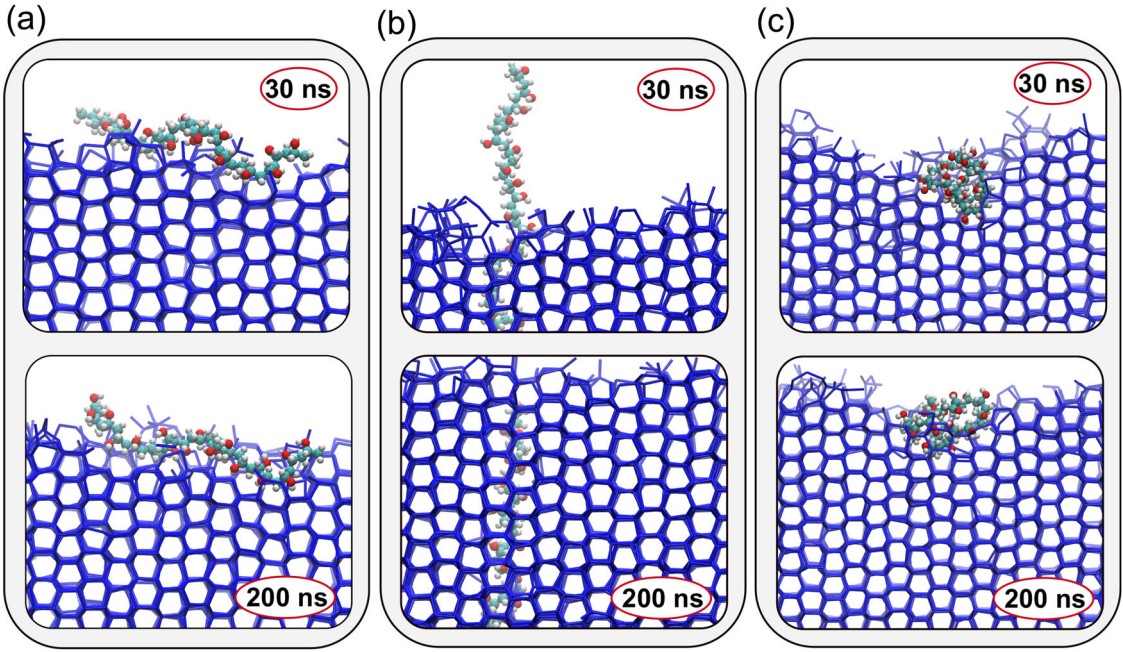

**Fig. 3 Lattice matching is not a prerequisite for PVA to bind to ice.** Representative snapshots of $PVA_{20}$ binding on a growing ice crystal. **a** An extended PVA chain ($R_g \sim 1$) is restrained so as to be perpendicular with respect to the growing ice front. **b** A similarly extended conformation is restrained so as to be parallel with respect to the growing ice front. **c** A compact, globular conformation ($R_g \sim 0.5$). All three conformations can bind ice, but only those illustrated in panels **a** and **c** show sustained IRI activity. Only water molecules belonging to the ice phase are shown.

characterising the pristine crystalline structure of the primary prismatic ice surface. As no simulations of the actual growth of the ice front were included (i.e. PVA bound on a static ice surface only) the roughness of the growing ice front was not taken into account. In contrast, in this work we have considered the PVA interactions with a dynamically growing ice front.

**IRI activity of PVA: ice-PVA contact area and occupied volume.** In order to explore the relationship between the chain length of the polymer and its IRI activity we have also simulated $PVA_{10}$ along with two identical $PVA_{10}$ polymers within the same volume of water (see Supplementary Fig. 6). Experimental results obtained via splat assay (see Congdon et al.[20]) have shown that a 1-mg/ml solution of $PVA_{10}$ lowers the mean largest grain size (MLGS) of ice crystals by ~20% if compared to a phosphate-buffered saline (PBS) control. In other words, $PVA_{10}$ displays some IRI activity: however, the same concentration of $PVA_{19}$ reduces the MLGS by ~80% – a much stronger impact on ice growth. The much weaker IRI activity of $PVA_{10}$ has often been attributed to the polymer not being able to bind ice well enough (or for a prolonged period of time) due to relatively low number of hydroxyl groups available to bind ice. To be specific, it has been suggested in the past that at least 6–8 units of PVA are needed for it to bind to ice[26] and recent computational work[24] has shown that there is a size dependence on the ice-binding efficiency of PVA – or, along the same lines, that the number of functional groups present on the chain dictates the degree of IRI activity [23]. Most recently, it has been proposed that $PVA_{10}$ might bind reversibly to ice, but lacks any IRI activity because the time bound to ice is shorter than the time it takes for ice to grow: thus the longer the PVA chain, the greater the chance of it to staying bound to ice and hence displaying IRI activity.[24].

While our results agree with previous findings that $PVA_{10}$ has reduced IRI activity, here we show that this is due to the polymer becoming overgrown once bound to ice, rather than not being able to bind for a sufficiently long enough time. Similar results

have been obtained for $PVA_{10}$ by Weng et al.[23]. In fact, out of the twenty MD simulations we have performed, $PVA_{10}$ bound to ice in nineteen of them: a representative example is illustrated in the left panel of Fig. 4. However, as opposed to $PVA_{20}$, $PVA_{10}$ is quickly overgrown, as shown in the middle panel of Fig. 4: in fact, we found that at mild supercooling (8 K) $PVA_{10}$ gets entirely embedded in the growing ice front within ~100 ns (see the right panel of Fig. 4). This can also be seen by tracking the number of PVA-ice hydrogen bonds over time (see Supplementary Fig. 5 as well as the Methods section).

Further insight into the weaker IRI activity of $PVA_{10}$ with respect to $PVA_{20}$ can be gained by monitoring the number of ice molecules as a function of time upon the binding of the polymer. This is shown in Fig. 5, where we have defined as $t_0$ the point in time at which either $PVA_{10}$ or $PVA_{20}$ bind to ice. The criteria for binding is met when 20% of the hydroxyl groups on the polymer are hydrogen bonded to the ice surface. Once $PVA_{20}$ binds to ice, the number of ice molecules plateaus, indicating a significant slow-down of the growth of the ice front; in fact, on the timescales probed via our simulations the ice front basically stops once it encounters $PVA_{20}$. Conversely, in the case of $PVA_{10}$, in 30% of cases we observe a temporary slow-down of the ice growth, followed by a sharp increase of the number of ice molecules which corresponds to the polymer becoming overgrown by the ice (see blue lines in graph). Importantly, these molecular-level details translate into a rather stark difference observed, experimentally, between the IRI activity of $PVA_{10}$ and $PVA_{20}$ – as shown in the right panels of Fig. 5. The two micro-graphs (obtained via splat assay, see Supplementary Methods) highlight the contrast between the mean grain size (MGS) of ice crystals growing in the presence of either $PVA_{20}$ or $PVA_{10}$: note the significant reduction of the MGS caused by $PVA_{20}$ in comparison to the much larger ice crystals observed for $PVA_{10}$.

We do observe, however, situations in which $PVA_{10}$ displays comparable IRI activity to that of $PVA_{20}$ (i.e. those lines not highlighted blue). This is consistent with experimental evidence[20] showing that $PVA_{10}$ does display limited IRI activity: for instance,

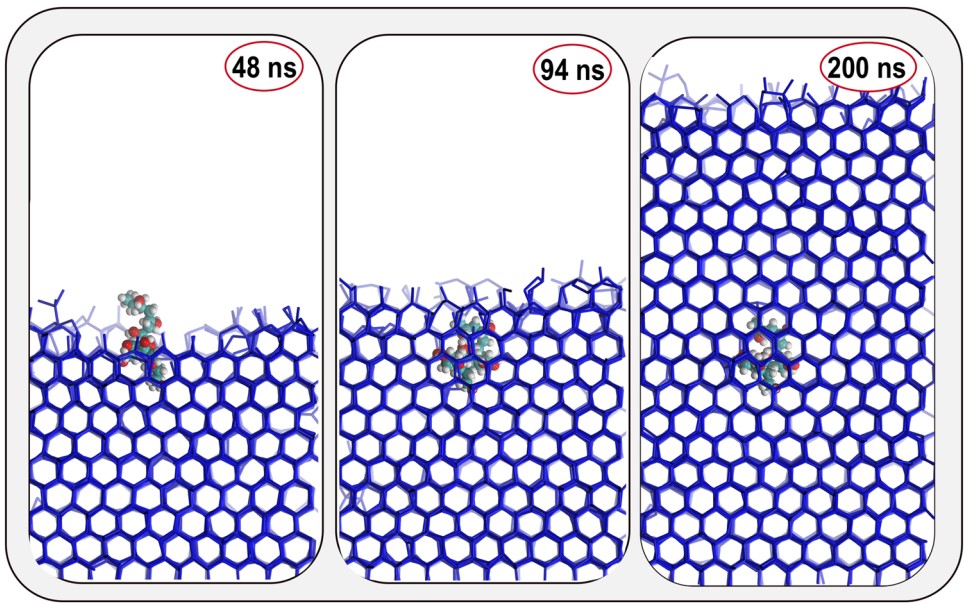

**Fig. 4 PVA$_{10}$ does bind to ice.** PVA$_{10}$ can bind to ice, independently of its conformation: however, the ice front overgrows the polymer due to the low DP of the latter, thus resulting in a lower IRI activity compared to e.g. PVA$_{20}$. Only water molecules belonging to the ice phase are shown.

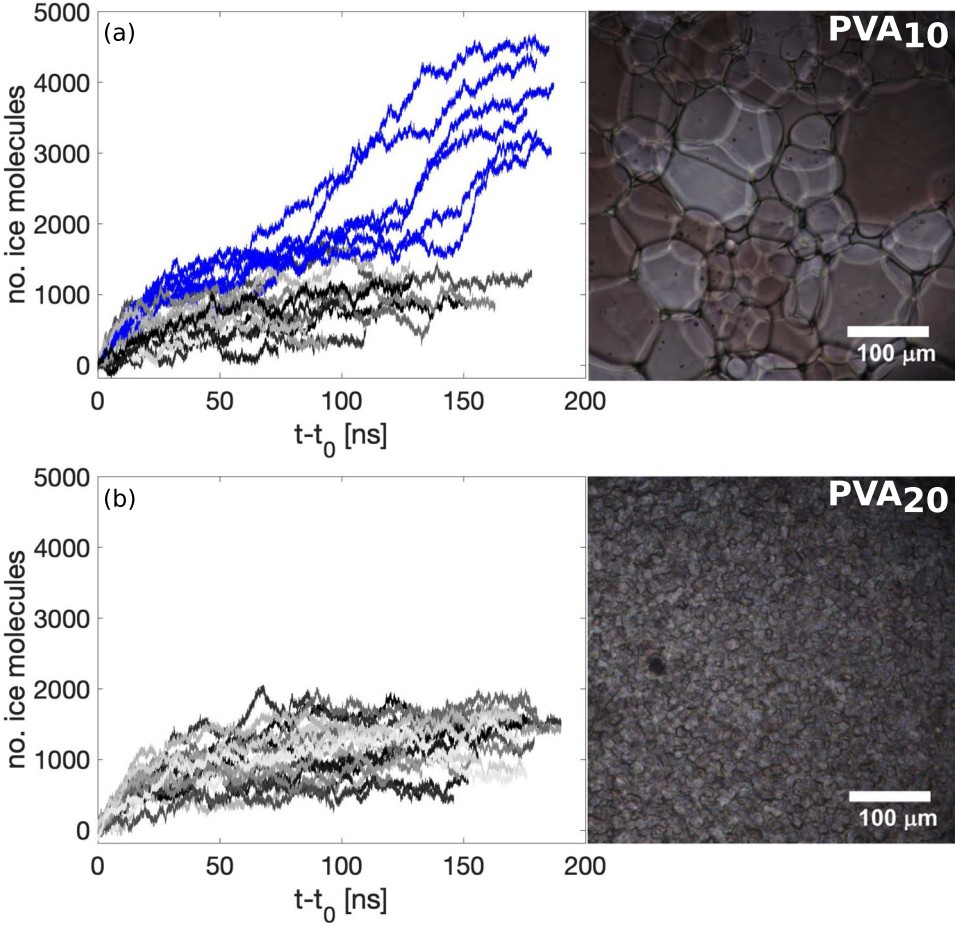

**Fig. 5 Comparison between the IRI activity of PVA$_{10}$ and PVA$_{20}$.** The IRI activity of PVA is quantified in terms of the evolution of the number of molecules within the growing ice front as a function of time. $t_0$ represents the time at which 20% of the hydroxyl groups on the polymer chain have bound to ice. The results of our MD simulations (left) are complemented by micro-graphs (right) obtained via splat assays experiments. **a** PVA$_{10}$: note the sharp increase in the number of ice molecules for those simulations (corresponding to the blue lines) in which PVA$_{10}$ becomes overgrown by the ice phase. **b** PVA$_{20}$: note the clear plateauing of the ice growth once PVA$_{20}$ has bound to the ice front. The results of the splat assays illustrate the dramatic difference in terms of the mean size of the ice crystals observes in the presence of either PVA$_{10}$ and PVA$_{20}$ – consistently with our MD simulations.

$PVA_{10}$ in concentrations between 2 and 5 mg/ml yields a MGS reduced by ~20–40% compared to PBS control. Thus, we argue that the IRI activity of $PVA_{10}$ is largely affected by the kinetics of ice growth, i.e., the time period during which $PVA_{10}$ manages to hold back the ice front before becoming engulfed in the ice phase might result in some IRI activity observed experimentally – but due to its eventual engulfment in ice its activity will be much weaker compared to $PVA_{20}$.

In light of these results, it is important to clarify the role of the "binding affinity" of PVA with respect to ice, which is often thought of as a linear function of the number of monomers[23,24,26]. As PVA tends to be found in the form of a random coil, the number of the monomers interacting with the ice surface does not increase linearly with the polymer length. Indeed, the binding affinities of $PVA_{10}$ and $PVA_{20}$, albeit difficult to quantify when dealing with dynamically growing ice surfaces, appear to be very similar.

Interestingly, we have found a correlation between the IRI activity of $PVA_{10}$ and the minimum distance (rMI hereafter) between the periodic images of the polymer. Specifically, $PVA_{10}$ polymers which became overgrown by the ice front are characterised, on average, by rMI = 3.7 ± 0.30 nm, while PVA10 polymers which showed some extent of IRI activity are characterised by an average value of 3.5 ± 0.30 nm. This trend holds for different surface areas as well; in particular, we have obtained, in the case of smaller simulation box (see Supplementary Fig. 10 for further details), rMI = 2.6 ± 0.16 nm and rMI = 2.3 ± 0.22 nm for inactive and active PVA10 polymers, respectively. These trends are indicative of the key role of the (polymer) surface coverage with respect to the (ice) surface – an aspect we discuss below.

We have also found that the length of the polymer chain has a stronger effect if compared to the concentration of the polymer in solution. In particular, within the same volume of water we have investigated the IRI activity of two $PVA_{10}$ polymers (which have the same molecular weight as one $PVA_{20}$). We have found that the growth of the ice front is largely unhindered by the two short-chain polymers, similarly to what we observe when considering a single chain of $PVA_{10}$ (see Supplementary Fig. 6 for further details). This conclusion holds even if we consider the statistically unlikely scenario by which the two $PVA_{10}$ polymers bind to the ice surface at exactly the same point in time. While we have not explored this possibility directly, we do know that a single $PVA_{10}$ interacting with an ice surface of ~26 nm² (which corresponds to the system discussed in this paper) shows the very same IRI activity as that of a single $PVA_{10}$ interacting with an ice surface of ~13 nm² (based on simulations carried out with smaller box boundary dimension and therefore smaller surface area, see Supplementary Fig. 10). This latter scenario effectively corresponds to two $PVA_{10}$ polymers interacting with an ice surface of ~26 nm² at the same time and thus serves to prove the point that indeed, two $PVA_{10}$ polymers are less effective in terms of IRI activity if compared to a single $PVA_{20}$. Lastly, this result is consistent with the experimental reality in that, as discussed in greater detail in the next sections, a tenfold increase in concentration is needed for $PVA_{10}$ to display an IRI activity similar to that of $PVA_{20}$, that is, at 10 mg/ml $PVA_{10}$ displays the same MGS of 30% as $PVA_{20}$ at 1 mg/ml. Thus, this evidence further strengthens the hypothesis that the number of functional groups binding to ice alone is not sufficient to trigger a substantial IRI activity.

Instead, we argue that it is the ice-PVA contact area, as well as the volume occupied by a bound PVA on ice, that rules the strength of its IRI activity. This proposition is illustrated in Fig. 6. We start by calculating, for $PVA_{10}$ and different conformations of $PVA_{20}$ as well, the number of ice molecules that grew within a time interval of 100 ns, starting from $t_0$. Note that each simulation is characterised by a different value of $t_0$, hence the reason for

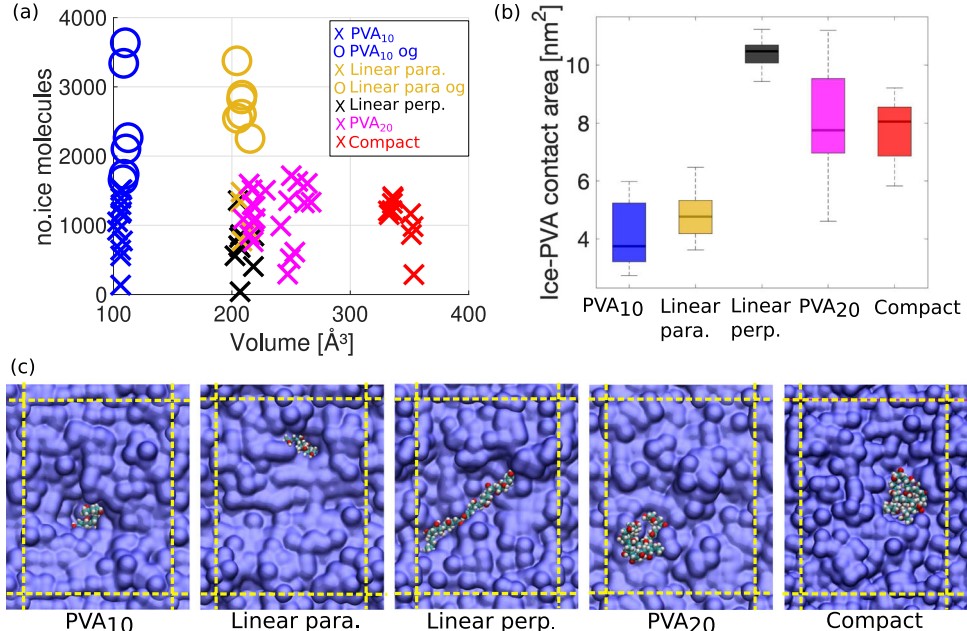

**Fig. 6 The IRI activity of PVA depends on its volume/surface area. a** Occupied volume of PVA on ice. IRI activity is measured by finding the number of ice molecules that make up the ice crystal at the end of the simulation (starting from the time a polymer binds to ice, $t$-$t_0$) and plotted against the average volume occupied by the polymers. "o" indicates a polymer has been overgrown (og) with ice, "×" indicates sustained IRI activity. Linear para. and linear perp. refer to the binding orientation to ice, parallel and perpendicular respectively. **b** Box-and-whisker plot illustrating the spread of the ice-PVA contact area. The upper(lower) bound and the line within each box represent the maximum(minimum) and the median. The extent of the boxes quantifies the upper and lower quartiles. **c** Representative cross-sections (xy-plane) of the system upon binding of PVA, the dotted yellow lines indicate the periodic boundaries.

choosing a specific time interval that allow us to compare the growth of the ice surface across different simulations. This measure of IRI activity is plotted in Fig. 6a as a function of the average volume occupied by PVA within the same time interval (see the Methods section for the details of the volume estimation). The lower the number of ice molecules we observe at the end of our simulation, the higher the IRI activity of PVA: circles in Fig. 6 corresponds to configurations which ended up overgrown by ice. $PVA_{20}$ restrained to a compact conformation ("compact" in Fig. 6) occupies, on average, the largest volume at the water–ice interface (~350 $Å^3$), followed by $PVA_{20}$ polymers free to adopt any conformation – and thus on average found to be as random coils; note the substantial spread of volumes, indicative of the variety of conformation that "unrestrained" PVA can assume when binding to ice. At lower volumes we find the "linearly restrained" polymers, i.e. extended, rod-like conformations that we force to be either perpendicular or parallel to the growing ice front ("Linear perp." and "Linear para." in Fig. 6, respectively). Lastly, we have $PVA_{10}$, occupying on average a much smaller volume (~110 $Å^3$) compared to any $PVA_{20}$ conformations. The random coil, compact and linear perp. conformations display the stronger IRI activity. On the other hand, the lower volume occupied by $PVA_{10}$ at the ice-water interface translates in a much weaker IRI activity. In addition, the linearly restrained $PVA_{20}$ conformations show another interesting trend: while linear perp. and para. are characterised by very similar occupied volumes, linear para. is much more likely to become overgrown by the ice phase, as illustrated in Fig. 3b. This is because the positioning of the linear para. conformation with respect to the ice surface minimises the interaction with the ice front – thus resulting in a much lower "effective" volume occupied by PVA at the ice–water interface compared to what we observe in the case of e.g. linear perp.

Overall, these results suggest that, once bound to ice, polymers occupying a small effective volume due to either their degree of polymerisation or their relative orientation with respect to the ice front will display similarly lower IRI activity. The binding of PVA and indeed, broadly speaking, of any polymeric IRI agent necessarily introduces a disruption of the crystalline order of the growing ice phase: small effective volumes such as those characterising $PVA_{10}$ or, equivalently, specific orientations of the polymer with respect to the ice surface (e.g. linear para. conformation), create "defects" in the ice phase which in some cases are clearly not extended enough to prevent the ice from growing further and eventually engulfing the polymer. (note by extension, this also shows that the number of functional groups binding to ice is not indicative of its IRI activity, as PVA chains overgrown by ice still display a significant number of hydrogen bonds, see e.g. Supplementary Fig. 3b).

We can further elaborate on the impact of the different binding orientations by considering the contact area between ice and PVA, which we have estimated according to the approach detailed in the Methods section. For instance, combining the information about volume (Fig. 6a) and contact area (Fig. 6b) we can understand the difference in terms of IRI activity between the linear para. and linear perp. orientations of the linear PVA conformation (see Fig. 6c). While both orientations are characterised by a very similar occupied volume, the parallel orientation spans a much smaller contact area compared to the perpendicular one. As a result, the parallel orientation is much more likely to get overgrown by the ice front, which results in a negligible IRI activity. We also note that the IRI activity of the different polymers/conformations shown in Fig. 6a is roughly the same if they sit above what we can identify as a minimum volume threshold: for instance, the linear perp. conformations are characterised by a smaller volume compared to "compact"

$PVA_{20}$, and yet they both show similar IRI activity. This result is consistent with a similar effect observed by Gibson et al.[32] in the context of the IRI activity of linear and star-branched PVA polymer. They showed that both architectures had similar IRI activity despite the star branch polymer having a much higher molecular weight, leading them to conclude that, due to the compact dimensions of the star-branched polymer, it is actually the hydrodynamic size rather than number of functional groups which is critical for IRI activity.

Thus, it is important to clarify that neither molecular weight nor the extent of the average contact area between any given IRI agent and the ice phase can be used, in isolation, to infer the IRI activity of polymers and anti-freeze proteins alike. Instead, it is the interplay between the volume and/or surface occupied by PVA and the ability of the latter to interact with the ice surface that dictates its overall efficiency as an IRI agent.

Finally, we also note that the activity of any given IRI agent strongly depends on its coverage of the ice surface: in this case, lower numbers of PVA polymers per (ice) unit area would lead to a lower IRI activity. This is the reason why, experimentally, a certain IRI agent displays activity above a certain concentration but it is not straightforward to translate (bulk) IRI agent concentration to the effective ice surface coverage. For completeness, we have investigated the IRI activity of both $PVA_{10}$ and $PVA_{20}$ as a function of different surface areas, namely 13.3 $nm^2$ (size S), 26.1 $nm^2$ (size M, the "main" size which we refer to throughout the paper), 53.4 $nm^2$ (size M/L) and 124.1 $nm^2$ (size L). The results are summarised in Supplementary Fig. 10 and suggest that $PVA_{20}$ is IRI active for surface coverages of 0.04 polymers/$nm^2$ or higher. In addition, the IRI activity of both $PVA_{20}$ and $PVA_{10}$ increases in the case of size S (compared to size M), albeit $PVA_{20}$ remain much more effective than $PVA_{10}$ even in this regime.

**PVA-ice interaction: enthalpic and entropic contributions**. It is clear from our simulations that the IRI activity of PVA relies on its ability of binding to ice. Here, we investigate the driving forces behind the PVA-ice interactions. To begin with, Fig. 7a shows the stark difference in terms of ice growth observed in the presence or absence of $PVA_{20}$. The system without PVA is used as a control: the advancement of the ice front (in Å, shaded green region in Fig. 7b) has been obtained as the average over 20 statistically independent MD simulations. Note that the finite extent of our water slabs prevents us from observing the expected linear regime of ice growth. To address this limitation we have thoroughly investigated the impact of finite size effect (see Supplementary Fig. 8): the setup we have chosen represents the best trade-off in terms of system size (~100,000 atoms as a whole) and computational resources and does not have an impact on the reliability of our results. Figure 7b also shows the growth of the ice front in the presence of (unrestrained) $PVA_{20}$ (dark green line, from a particular simulation); note the sharp decrease in the growth rate due to the binding of $PVA_{20}$ to ice (shown in blue).

The driving force behind PVA binding to ice is widely accepted to be enthalpically driven by the formation of hydrogen bonds. In Fig. 7c we report the time evolution of the fraction of the PVA hydroxyl groups that are hydrogen bonded to the ice surface. We show fluctuations between 20 and 60%, with a moving average (black line in Fig. 7c) of ~35%. The extent of these fluctuations are due to the ever-changing roughness of the growing ice front as well as to the mobility of the polymer – particularly those segments not bound to the ice surface. Previous computational work[22] has suggested that the PVA-ice-binding arises from a linear, extended arrangement of the polymer which should be lying flat on top of the ice: if we take into account the 2:1 binding

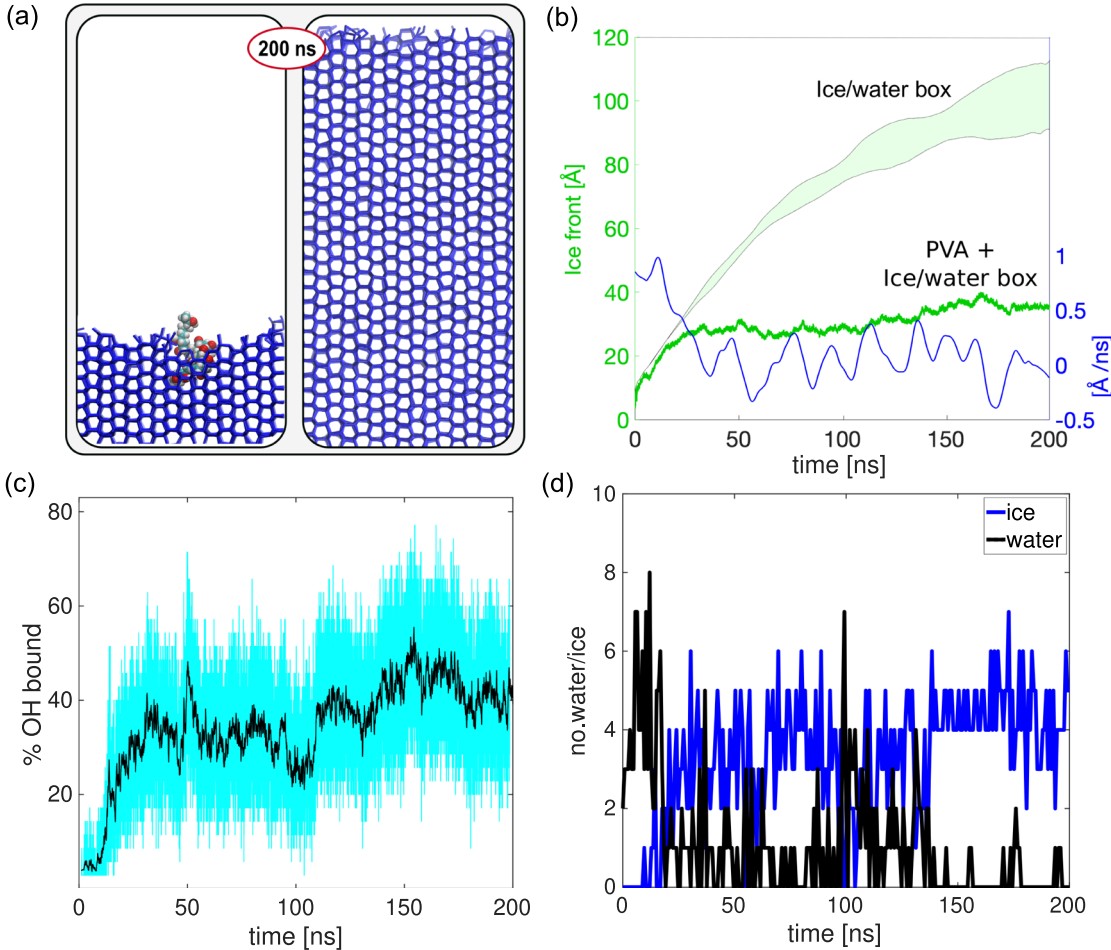

**Fig. 7 Hydrogen bonding and hydrophobic interactions. a** Comparison of ice growth in the presence (left inset) and absence (right inset) of PVA$_{20}$.
**b** Advancement of the ice front (in Å) for the ice/water systems only (the shaded green region represent the spread of our data with reference to twenty independent MD simulations) and in the presence of PVA - single green line, referring to the particular simulation illustrated in panel **a**; the growth rate of the ice phase for the same PVA-inclusive simulation is also shown (single blue line). **c** Time evolution of the percentage of hydroxyl groups forming hydrogen bonds (light blue): the black line corresponds to the moving average. **d** Representative time evolution of the "type" of water molecules (either liquid-like or ice-like, see Methods section) found within the first solvation shell of a PVA methylene group interacting with the ice surface.

pattern also discussed in Naullage et al.[22], we should observe, on average, ~70% of hydroxyl groups forming hydrogen bonds to ice. However, our results (which hold for all the simulations see Supplementary Fig. 4) suggest instead that a much lower fraction of hydroxyl groups is needed for PVA to effectively bind to ice. This evidence is consistent with the conformational arguments we have discussed above: given that PVA can typically be found in a random coil conformation, only a fraction of its hydroxyl group is in a position to interact with the growing ice front at any given time. We therefore suggest additional driving forces may contribute to PVA-ice binding, other than the enthalpic gain from hydrogen bonding.

At this stage, it is interesting to consider the possibility that the PVA-ice interaction might benefit from a small additional entropic gain. This hypothesis is motivated by the ever-growing evidence suggesting that hydrophobic interactions play a pivotal role in the context of IRI activity (see e.g. refs. [33–36]). Thus, we have investigated the desolvation of the methylene (-CH2) hydrophobic groups of PVA, which might lead to an entropic contribution in addition to the enthalpic gain originating from the PVA-ice hydrogen bonding. In solution, the first solvation shell of the methylene groups contains on average five water molecules (see Supplementary Fig. 1 as well as the Methods

section). The desolvation of methylene groups upon their interaction with ice is illustrated in Fig. 7 and corresponds to the drop in the number of water molecules within the solvation shell and the corresponding increase of ice molecules. As the solvation shell of methylene is structurally constrained compared to the bulk liquid phase, there is an entropic contribution associated with the desolvation of the -CH2 groups.

To probe this possibility in greater detail, we have investigated the IRI activity of polypropylene (PP), an entirely hydrophobic polymer. While PP is in reality insoluble in water, this computational experiment allows us to probe the potential effect of the desolvation of methylene groups in absence of hydrogen bonding. We find that PP does indeed interact with the growing ice front via purely hydrophobic interactions originating from the desolvation of methylene groups (see Supplementary Fig. 9). Unlike PVA, PP does not stop the growth of the ice front but it can slow it down somewhat. This mechanism is identical to that reported for PVA in Fig. 7 and it is consistent with the work of Naullage et al.[22] which also suggests the existence of a small entropic contribution due to the hydrophobic desolvation of PVA methylene groups. In the attempt to unravel the interplay between enthalpic and entropic contributions, we have also investigated the IRI activity of polyvinylamine (PVAm, DP = 20),

via MD simulations as well as splat-assay measurements, summarised in Fig. 8. We find the IRI activity of PVAm to be substantially lower than that of PVA.

The main difference between PVA and PVAm is that in PVAm the hydroxyl groups are replaced by primary protonated amines. The latter can form hydrogen bonds but not as readily as hydroxyl groups. This is due to the asymmetric nature of the hydrogen bonding in protonated amines: that is, they are more likely to offer donors, rather than acceptors[37]. Furthermore, the PVAm hydration shell structure is likely to be affected by the net positive charge of the amine group, which contributes to explain why we observe only minimal interaction with the growing ice front for PVAm compared to PVA. In turn this indicates that the predominant driving force for the binding of PVA to ice is mostly enthalpic in nature, i.e., comes from its hydroxyl groups. Overall, we are in no position to quantify this subtle balance between enthalpy and entropy. However, in light of all of the above we can confidently put forward that entropic contributions can play a role in the context of IRI activity. This is important not just in the context of IRI-active polymers but to, e.g., amphiphilic ice-binding proteins and other systems as well.

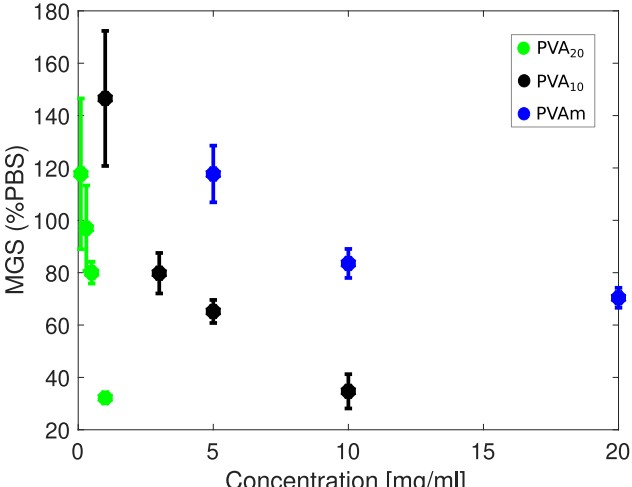

**Fig. 8 Experimental IRI activity of PVA$_{20}$, PVA$_{10}$ and PVAm.** Mean grain size (MGS) of the ice crystals measured via splat-assay in the presence of different polymers, as a function of their concentration. The error bars represent the standard deviation for each sets of measurements. The results have been normalised relative to a PBS control. Our MD simulations are consistent with experimental results which show that PVA$_{20}$ is the most IRI active, with PVA$_{10}$ and PVAm showing far less activity even at higher concentrations. Note that a 30% MGS can be obtained for PVA$_{20}$ at a concentration of just 1 mg/ml, while a tenfold concentration increase would be required for PVA$_{10}$ to display a similar activity. The molecular weight of PVAm (DP = 580) is 25000 g/mol.

**The IRI activity of PVA-$b$-PVAm block copolymers.** In light of our findings which show that the low IRI activity of PVA$_{10}$ is largely dictated by its small effective volume/contact area at the ice-water interface – resulting in its engulfment – we have been prompted to investigate the effect on PVA's IRI activity when substituting some of the hydroxyl groups with groups less likely to form hydrogen bonds to ice. To this end, we have focused on PVAm, which is characterised by protonated primary amines (-NH$_3^+$). The splat-assay results reported in Fig. 8 for PVAm (DP = 580) indicate that this polymer is rather ineffective as an IRI agent even at really high concentrations. Indeed, our MD simulations show that this is because PVAm simply does not bind to ice during 20 statistically independent simulations (200 ns long each).

It is interesting to observe what happens when only a fraction of the PVA's -OH groups are substituted with -NH$_3^+$ fragments: in particular, we have studied three different block copolymers, two with DP = 20 (PVA$_{13}$-$b$-PVAm$_7$ and PVA$_5$-$b$-PVAm$_{15}$) and one with DP = 10 (PVA$_5$-$b$-PVAm$_5$). As illustrated in Fig. 9, these block copolymers were all able to bind ice and slow down the growth of the ice front (see Supplementary Fig. 7 for further details). Consistent with what we have observed in the case of PVA, here we also show that the lattice matching argument does not hold. In fact, as shown in Fig. 10 the majority of our simulations show that it is the terminal end (i.e. the PVA units) of the copolymers that hydrogen bond to ice in a conformation parallel with respect to the ice front – the NH$_3^+$ segments are hardly involved in ice interactions.

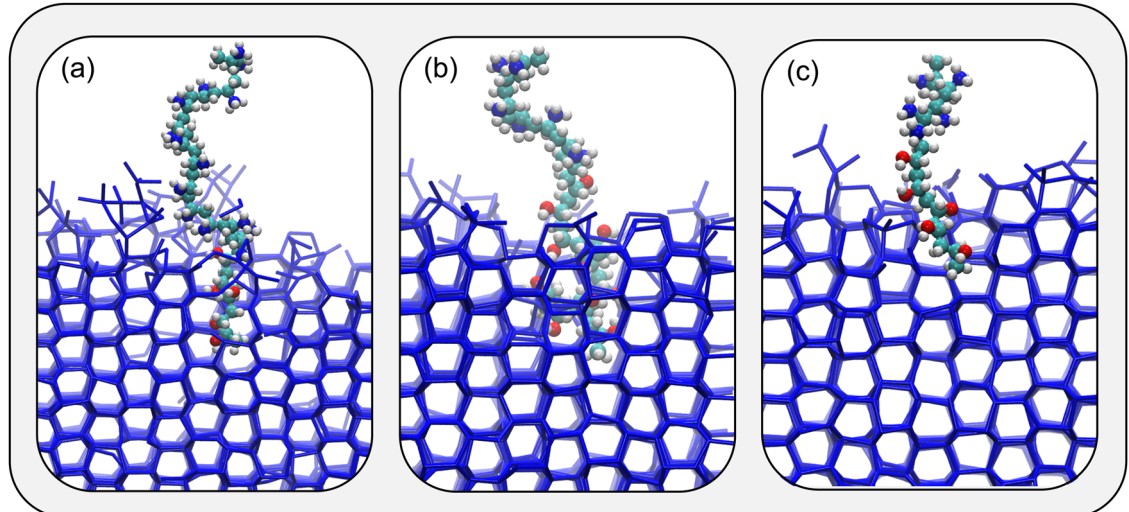

**Fig. 9 Representative snapshots of IRI-active PVA copolymers binding to ice. a** PVA$_{13}$-$b$-PVAm$_7$. **b** PVA$_{13}$-$b$-PVAm$_7$. **c** PVA$_5$-$b$-PVAm$_5$. The PVA segments can be identified via the hydroxyl groups (oxygen and hydrogen atoms are show in red and white, respectively), while the PVAm segments correspond to the amine functional groups (blue spheres repesents nitrogen atoms). The polymers typically bind to ice via the PVA segments only. Only water molecules belonging to the ice phase are shown.

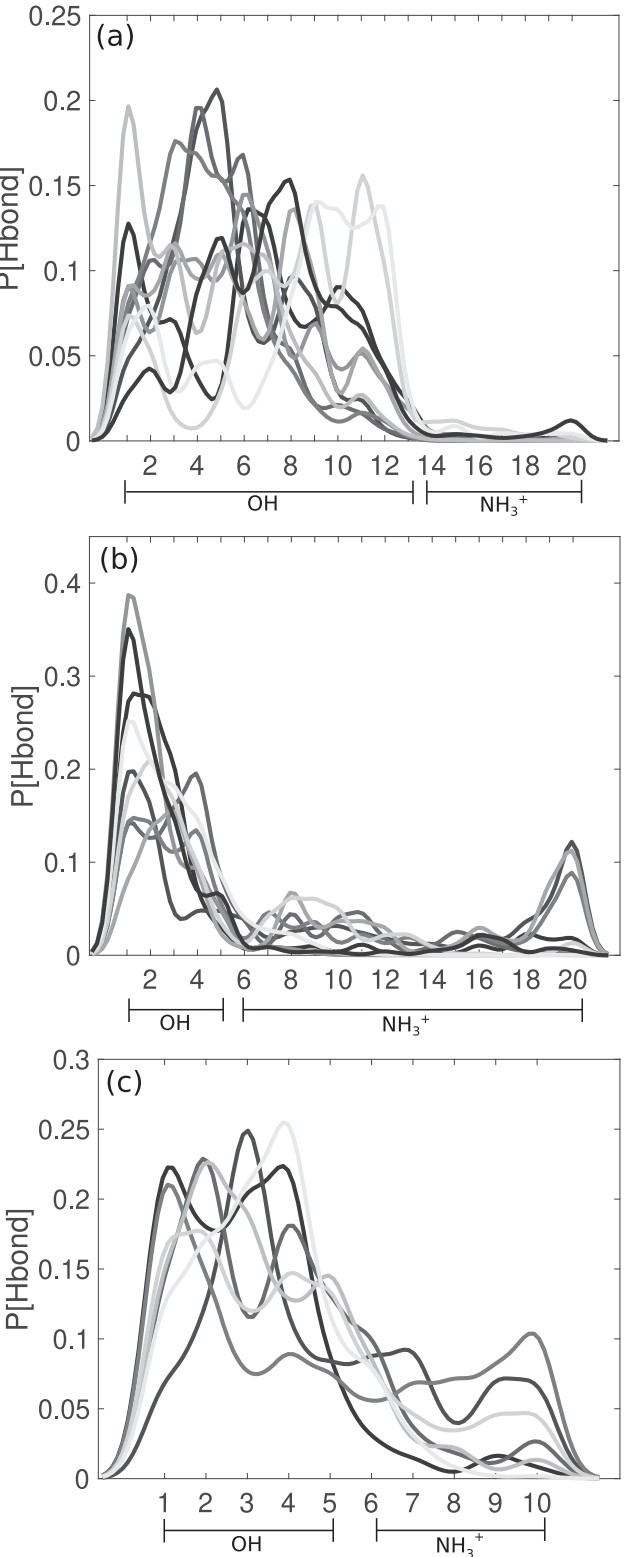

**Fig. 10 Hydrogen bonding between PVA-b-PVAm block copolymers and ice. a** Probability density functions of the number of hydrogen bonds between the -OH or -NH$_3^+$ functional groups and the ice surface in the case of PVA$_{13}$-b-PVAm$_7$. **b** Same as **a** but for PVA$_5$-b-PVAm$_{15}$. **c** Same as **a** but for PVA$_5$-b-PVAm$_5$. The results have been averaged over 10 statistical independent MD simulations.

Previous experimental work[38] on PVA-b-PVP block copolymers has shown that the addition of PVP (polyvinylpyrrolidone) had little influence on the IRI activity, with the block polymers presenting roughly the same activity levels as the PVA homopolymers. In particular, a range of PVA$_{10}$-b-PVP$_n$ block copolymers were examined where the MLGS of PVA$_{10}$ homopolymer was roughly the same as that of the block copolymers. Outside the realms of synthetic polymers, it has also been shown[39] that a particular 7 kDa anti-freeze protein (AFP) has identical IRI activity to a 7 kDa AFP with a further 42 kDa protein appended to it. Hence, both studies suggest no increase in activity by appending further monomeric units or protein segments to the initial starting structure.

At this stage we would like to point out that, as opposed to the previous experimental works discussed above, here we are not adding extra units to the polymer: rather, we wish to assess whether the IRI activity of block copolymers (with DP of either 10 or 20) can be affected by substituting some of their hydrogen bonding functional groups with -NH$_3^+$. We also note that while our MD simulations take into account a well-defined DP, the vast majority of synthetic polymers investigated in the context of IRI activity are typically polydisperse mixtures encompassing a range of DP. This aspect was investigated by Gibson et al.[40], who showed that even a low dispersity mixture (with a dispersity index of e.g. 1.2), obtained via RAFT polymerisation, can contain small amounts of high molecular weight polymers which dominate the IRI activity, thus yielding false positives in terms of the IRI activity of low molecular weight PVA. Ultra-low dispersity indexes (as low as 1.005) could be obtained via column chromatography: the resulting polymers indicate that a DP = 12 represents the minimum value required to observe a significant IRI activity – albeit at much higher concentrations than PVA$_{20}$, i.e., to observe similar MLGS for 5 mg/ml of PVA$_{20}$, 25 mg/ml of PVA$_{12}$ would be needed.

In contrast, our results indicate that even the short block copolymer PVA$_5$-b-PVAm$_5$ (DP = 10, <12) is capable of slowing down the growth of the ice front as shown in Supplementary Fig. 7, despite the fact that its occupied volume is similar to that of PVA$_{10}$. The non-negligible IRI activity of PVA$_5$-b-PVAm$_5$ compared to PVA$_{10}$ originates from the fact that the primary protonated amines such as the NH$_3^+$ groups characterising the PVAm segment of PVA$_5$-b-PVAm$_5$ form fewer hydrogen bonds compared to hydroxyl groups (as discussed above in the case of PVAm). As such, it is much more difficult for the growing ice front to form hydrogen bonds with the PVAm section of the block copolymer, see Fig. 10 – which in turn prevents the ice from overgrowing the polymer.

Interestingly, the PVA$_{13}$-b-PVAm$_7$ copolymer (which has the same DP as PVA$_{20}$) displays similar IRI activity compared to the unrestrained PVA$_{20}$ polymer (compare Fig. 5 with Supplementary Fig. 7a) and our MD simulations show this copolymer binding to ice. This result provides further evidence that it is not necessary for the entire polymer chain to bind to ice in order for the polymer to display substantial IRI activity. This is clearly highlighted in Fig. 10a, where only 13 units out of a 20 unit long chain have a high probability of binding to ice. Moreover, PVA$_5$-b-PVAm$_{15}$ and PVA$_5$-b-PVAm$_5$ show similar IRI activity (see Supplementary Fig. 7b, c). Thus, it seems that even a short (5 monomer) PVA segment might be able to bind to ice and show IRI activity – provided the rest of the polymer chain features functional groups that prevent the engulfment of the entire polymer into the growing ice phase. To this end, bulky functional

groups with only limited potential to hydrogen bond might be most effective, while – as discussed in the previous section – entirely hydrophobic group can trigger favourable polymer-ice interactions.

This is a particularly interesting finding of direct practical relevance, as our results suggest that it might be possible to enhance the IRI activity of e.g. PVA$_{10}$ by including specifically functionalized segments. Thus, we argue that, in light of our results, there is some potential to develop small synthetic polymers that can be engineered to have the same IRI activity as PVA$_{20}$ but with a potential DP = 10. This would represent a major step forward in the context of cryopreservation, as the attachment of smaller polymers to ice (relative to larger ones) lowers the probability of observing potential side effects associated with supercooling beyond the thermal hysteresis gap, such as dynamic ice shaping (see refs. [26,41]). We do however, note that 3/10 simulations for PVA$_5$-$b$-PVAm$_{15}$ and 2/10 simulations for PVA$_5$-$b$-PVAm$_5$ showed no binding activity, whereas the copolymers with 13 units of PVA (PVA$_{13}$-$b$-PVAm$_7$) bound to ice in all simulations (see Supplementary Fig. 7). There is therefore a potential size dependency for very small segments of PVA (DP < 10) binding ice, as mentioned in Naullage et al.[24] We hope that our results will encourage future studies which systematically investigate the existence of an optimal ratio between ice-binding and non-ice-binding segments for a given type of block copolymer which could lead to small IRI active polymers.

## Discussion

Ice recrystallisation is a real challenge within the field of cryobiology, as it leads to irreversible damage of biologically frozen samples – this is both wasteful of precious resources as well as costly. Understanding how and why a cryoprotective agent such as PVA works is key to move forward with the development of synthetically accessible IRI active compounds.

In this work, we have used all-atom molecular dynamics simulations to gain a microscopic understanding of the PVA-ice interaction. We have found that, in contrast to the current thinking, the lattice matching between the functional groups of PVA and the ice does not play a role. This is because PVA is a very flexible polymer which can bind to ice in any conformation, as we have shown via an unprecedented combination of metadynamics and (un)constrained MD simulations. This is a very important result, as it removes a long-standing design constraint, that of lattice matching, from the rational design of IRI active polymers. It is important to stress that lattice matching might be playing an important role for different classes of polymers and/or different types of IRI agents. In particular, the recent body of work on ice-binding proteins[33,35,42] suggest that lattice matching is key when dealing with large and rigid IRI agents, as opposed to small, flexible polymers like PVA$_{10}$ and PVA$_{20}$.

Furthermore, we have showed that the weaker IRI activity of short polymers (such as PVA$_{10}$ compared to PVA$_{20}$) is due to the shorter polymer becoming overgrown by the advancing ice front, rather than – as previously thought – the limited number of hydroxyl groups binding to ice. In fact, PVA$_{10}$ does bind to ice and we have established that the IRI activity of PVA$_{10}$ is lower than that of PVA$_{20}$ because the former occupies a much smaller effective volume/surface than the latter at the ice-water interface. These results are particularly interesting as they provide an explanation for the lower – and yet non-negligible – IRI activity of PVA$_{10}$ compared to longer polymers.

In investigating the driving forces responsible for the PVA-ice interaction, we have found that, in addition to the enthalpy gain originating from the hydrogen bonds between PVA and ice,

entropic contributions due to the desolvation of methylene groups can also play a role. While the interplay between enthalpy and entropy is probably skewed toward the former in the specific case of PVA, we argue that such a competition has to be taken into account when designing a potential cryoprotectant.

Lastly, we have probed the IRI activity of PVA-$b$-PVAm block copolymers: we have demonstrated that it is not necessary for the whole polymer chain to bind ice in order to observe IRI activity. Indeed, we have found that even small chain block copolymers of PVA$_5$-$b$-PVAm$_5$ (i.e. DP = 10) have the potential to exhibit significant IRI activity. This finding is of great relevance in the context of the rational design of novel cryoprotective agents, as it shows that it is possible to synthesise very short (DP < 20) IRI active polymers. From a practical standpoint, this is especially intriguing, as smaller cryoprotectants are less likely to cause dynamic ice shaping – a potentially lethal occurrence which can exacerbate the problem of cryopreservation.

We believe these findings are not only of great relevance to cryobiology and medical applications, but they are bound to make an impact on many other areas where ice recrystallisation plays a role, such as the development of freeze resistant crops and the food industry – as well as furthering our fundamental knowledge within the realms of atmospheric sciences and climate change. We are also confident that our results will foster future experimental and computational works aimed at probing the IRI activity of more complex copolymers, thus laying the foundations toward the rational design of the next generation of cryoprotectants.

## Methods

**Molecular dynamics simulations**. All the molecular dynamics simulations reported in this work have been preformed using the all-atomistic CHARMM36 forcefield[43] along with the TIP4P/Ice water model[44], using the MD package GROMACS 5.1.3[45]. There is a large body of evidence indicating that the CHARMM36 - TIP4P/Ice combination of force fields is especially well suited to perform MD simulations of biomolecules in supercooled water (and in contact with ice)[34,43,44,46–48]. In our simulations, we have investigated the atactic poly (vinyl)alcohol (PVA) polymer with various degree of polymerisation (PVA$_{20}$, PVA$_{10}$) and conformational flexibility. We have found that the tacticity of PVA does not have an impact on our results. We have also simulated a range of poly-vinylalcohol/polyvinylamine block copolymers (PVA-$b$-PVAm) characterised by different ratios of hydroxyl to amine functionalisation.

For the system setup shown in Fig. 1, we have ran 20 independent simulations, each one 200-ns long, for PVA$_{20}$, PVA$_{10}$ and 2xPVA$_{10}$ polymers (see Supplementary Fig. 6 for further details about the latter), 10 simulations, each one 200 ns long, where the conformation of PVA$_{20}$ was restrained to either a compact or linear geometry (see Fig. 1), and 10 simulations, each one 200 ns long, for three different PVA-$b$-PVAm block copolymers. We also ran 20 simulations, each one 200 ns long, of the ice/water system in absence of any polymer, so as to have reference/control benchmark.

To begin with, the geometry of a given polymer was optimised using a steepest descent algorithm[49]. Then, each polymer was solvated in water within a cubic simulation box (edge = 7.8 nm) and equilibrated for 30 ns at room temperature and ambient pressure within the NPT ensemble: to this end, we have employed the Bussi-Donadio-Parrinello thermostat[50] and the Berendsen barostat[51], with coupling constants of 0.5 and 4 ps, respectively. Periodic boundary conditions (PBCs) were applied in three dimensions and the integration time-step for the leap-frog algorithm was set to 2 fs. In parallel, a separate ice/water box was created, starting by preparing an ice crystal (of dimensions 4.7 × 5.9 × 3.9 nm) cleaved so as to have its primary prismatic face lying in the xy-plane of the simulation box: this crystal acted as the seed for the ice phase to grow along the ±z direction (see Fig. 1). The experimental evidence reported by Budke and Koop Due (see Budke and Koop[26]) demonstrates that PVA chiefly interacts with the primary prismatic face of hexagonal ice. Thus, in this work we have chosen to focus on this particular surface, in line with previous computational work[22,23].

The oxygen atoms within the ice slab were subjected to position restraints via a harmonic potential characterised by a spring constant of 10,000 kJ/mol; the system was then solvated, adding two layers (each one ~11 nm thick) of water below and above – a total of 90,818 atoms. The simulation box was elongated along the z direction to include ~22.5 nm of vacuum: at this point we switched to 2D (xy) PBCs in conjunction with 9-3 Lennard-Jones "walls"[52] positioned at the top and bottom of the box. This strategy enabled us to use the Yeh and Berkowitz correction term[53] to the Ewald summation: in turn, this approach mitigates any potential artefacts due to the treatment of electrostatic interactions when dealing

with a slab geometry[54]. The cutoff for the van der Waals and electrostatic interactions was set to 12 Å and 10 Å, respectively: a switching function was used to bring the van der Waals interaction to zero at 12 Å. The geometry of the water molecules was constrained using the SETTLE algorithm[55] while the P-LINCS algorithm[56] was used to constrain the hydrogen bonds for the polymers at their equilibrium value. This setup has been extensively validated in some of our previous works, see e.g. Sosso et al.[48]. The ice/water system was subsequently equilibrated (NPT ensemble, using a semi-isotropic pressure coupling scheme) at 300 K for 50 ns. Once the polymer and the ice/water system were independently equilibrated, two polymers were then placed into the ice/water box within the two water slabs, one above and one below, as per Fig. 1. The system was then further equilibrated for 20 ns at 300 K and then cooled down to 265 K within 10 ns: the polymers were not restrained so during equilibration they were free to diffuse the entire water slab. A 200-ns production run at 265 K followed, switching to the Parrinello-Rahman barostat[57] with a coupling constant of 4 ps. The starting configurations of PVA in the water slab depended on the last frame in the equilibration run and as such all trajectories had independent starting configurations. To prevent the polymers interacting with the water/vacuum boundary, a constraint for the centre of mass of the polymer was set to be at least 1 nm below the water-vacuum boundary layer. Configurations were collected every 2000 MD steps.

When dealing with simulations of crystal nucleation and growth, the impact of finite size effects can be especially significant[58]: as such, we have conducted a systematic investigation, discussed in detail in the Supplementary Information (see Supplementary Fig. 8). We note that the ice recrystallisation process involves the coalescence of different ice crystals embedded in supercooled liquid water. Thus, it is appropriate to address the interaction between PVA and ice at the interface between the latter and supercooled liquid water. While our simulations provide an indirect assessment of the IRI activity of PVA (in that we are not simulating the coalescence of different ice crystals), the interaction between PVA and ice is ultimately the molecular-level mechanism responsible for the extent of ice recrystallisation. In fact, our simulations are in excellent agreement with the experimental splat assays (which utilise NaCl to ensure an equilibrium between the ice crystals and the supercooled liquid phase) discussed in the results section of this paper.

**Identifying icy molecules and hydrogen bonds.** To determine whether a certain water molecule belongs to the ice crystal or the supercooled liquid phase, we have employed a strategy based on the Steinhardt order parameters[59]. First, we compute the 6th order Steinhardt vectors $q_{6,m}(i)$ as:

$$q_{6,m}(i) = \frac{\sum_{j \neq i}^{N} \sigma(|\mathbf{r}_{ij}|) Y_{6,m}(\mathbf{r}_{ij})}{\sum_{j \neq i}^{N} \sigma(|\mathbf{r}_{ij}|)}, \tag{1}$$

where $\mathbf{r}_{ij}$ is the distance vector between the $i^{th}$ and $j^{th}$ atom, $Y_{6,m}$ is a spherical harmonic of order $\{6, m\}$ and $\sigma$ is a switching function which determines the extent of the of coordination shell. Then, we combine these Steinhardt vectors to obtain the following order parameter:

$$s_6(i) = \frac{\sum_{j \neq i}^{N} \sigma(|\mathbf{r}_{ij}|) \sum_{m=-6}^{6} q_{6,m}^{*}(i) \cdot q_{6,m}(j)}{\sum_{j \neq i}^{N} \sigma(|\mathbf{r}_{ij}|)} \tag{2}$$

where the asterisk denotes complex conjugation. By means of the clustering algorithm described in Tribello et al.[59], we identify the largest connected cluster of water molecules which oxygen atoms display a value of $s_6(i)$ greater than a certain threshold (0.45, see Tribello et al.[59] for further details). Thus, the result of this procedure provides the number of water molecules found within the largest ice cluster per trajectory frame, which invariably will be the seeded ice crystal that grows over time. Concerning the analysis of hydrogen bonds, we have used a geometric criterion based on an acceptor-donor bond distance of <0.3 nm and a donor-hydrogen-acceptor angle between 160° and 200°. The analysis on the solvation shell around the methylene groups was based on the first minimum of the radial distribution function ($g(r)$ see Supplementary Fig. 1) where a radial distance of 0.45 nm corresponds to the extent of the first solvation shell. Using the $s_6$ parameter described above we were able to distinguish which water molecules surrounding the methylene groups can be classified as either ice-like or liquid-like.

**Metadynamics simulations.** To probe the conformational space of the PVA polymer we use well-tempered metadynamics[60,61], an enhanced sampling technique which forces the system to explore the whole free energy surface relative to a particular set of collective variables (CVs). This is achieved by depositing Gaussian potentials $V$ at a defined time interval, $t$:

$$V(s, t) = \sum_{k\tau} W(k\tau) \exp\left(-\frac{s - s(q(k\tau))^2}{2\sigma^2}\right) \tag{3}$$

where $s$ is a given CV, $\tau$ is the Gaussian deposition stride and $\sigma$ the width of the Gaussian for the $i$th CV and $W(k\tau)$ the height of said Gaussian potential. We have used PLUMED[62,63] (version 2.4.2) to perform metadynamics simulations. We have chosen the radius of gyration ($R_g$) as our CV to bias: the width, height and deposition stride of the Gaussian potentials has been set (after extensive testing and

validation) to $\sigma = 0.02$ nm, $W = 0.6$ kJ/mol, and 500 steps, respectively. The bias factor was set to $\gamma = 100$. The uncertainty associated with our estimate of the free energy profile reported in, e.g., Fig. 2 was calculated using the reweighing technique of Tiwary and Parrinello[64]. We have also used PLUMED to perform the MD simulations where the polymers conformation was restrained (to either their compact or elongated forms, see Fig. 3): specifically, the value of the polymers' radius of gyration ($R_g$) was enforced to fluctuate around values of 4.9 and 1.2 nm in the case of a compact and elongated conformation, respectively.

**Estimating molecular volume and ice-PVA contact area.** To measure the average volume occupied as well as the surface area for a given polymer we employed the concept of alpha shape[65]: by treating the atoms within the polymer as a set of points in the Euclidean space, the alphaShape function implemented in MATLAB[66] identifies the smallest shape enclosing said atoms – which in turn gives us access to the volume of the molecule as a whole as well as the surface area of the polymer occupied on the ice slab (obtained by projecting the atomic positions of the polymer bound to the ice onto the xy plane). The key parameter in this procedure is the so-called alpha radius, which basically determines how loose or tight the fit of the alpha shape is around the set of points under investigations: we have set this value to 0.17 nm, but we note that different choices do not affect the qualitative trends we report in e.g. Fig. 6.

**Ice recrystallisation inhibition (Splat) assay.** A 10 μL sample of polymer was dissolved in phosphate-buffered saline (PBS, known to display zero IRI activity, pH 7.4) and then dropped from 1.40 m onto a glass microscope cover slip, which sat on top of an aluminium plate cooled to −78° using dry ice. The droplet froze instantly upon impact with the plate, spreading out and forming a thin wafer of ice. This wafer was then placed on a liquid nitrogen cooled cryostage held at −8°. The wafer was then left to anneal for 30 min at −8°. Next, the number of crystals in the image were counted using ImageJ[67], and the area of the field of view divided by this number of crystals gave the average crystal size per wafer (mean grain size, MGS hereafter), and reported as a percentage of area compared to the PBS control. We note that along with MGS another quantity MLGS (mean largest grain size) can be computed which is the largest grain in the field of view and therefore slightly biased towards lower activity, however this quantity can be measured without image analysis software. For further details about the physical, analytical and synthesis methods, please refer to the Experimental Section in the SI.

## Data availability

The research data supporting this publication can be found at http://wrap.warwick.ac.uk. The portfolio of codes we have used to generate and analyse our results includes: GROMACS[45], PLUMED[62,63], MDAnalysis[68] and MATLAB[66]. These are all well-documented packages that are all publicly available. MATLAB does require a license, which is however covered by the vast majority of academic institutions.

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

## Acknowledgements

We gratefully acknowledge the use of the ARCHER UK National Supercomputing Service (http://www.archer.ac.uk), which we have accessed via the HecBioSim consortium (funded by the EPSRC grant no. EP/R029407/1). We also acknowledge the use of Athena at HPC Midlands+, which was funded by the EPSRC via the grant n.EP/P020232/1, via the HPC Midlands+ consortium. We gratefully acknowledge the high-performance computing facilities provided by the Scientific Computing Research Technology Platform at the University of Warwick. We would also like to acknowledge the research project grant from the Leverhulme Trust (RPG-144), as well as the ERC (638661) and the Royal Society for the Industry Fellowship (191037).

## Author contributions

F.B. performed the simulations and analysis. T.R.C. and C.S performed the experiments and analysis. F.B., G.C.S. and M.I.G. interpreted the results. G.C.S and M.I.G. conceived the research. F.B. and G.C.S. wrote the manuscript.

## Competing interests

The authors declare no competing interests.
