## [Peer Review File · Nature Communications]

REVIEWER COMMENTS

Reviewer #1 (Remarks to the Author):

This paper reports molecular dynamic (MD) simulations aimed at investigating the ice recrystallisation inhibition (IRI) activity of poly(vinyl)alcohol (PVA). Specifically, using atomistic models for water and PVA in their simulations, the authors systematically probe the impact of conformation, number of functional groups, and degree of polymerisation. In contrast to previous studies, they find that PVA can bind ice in any conformation and that lattice matching is not a prerequisite for ice inhibition. They also show that short-chain PVA's reduced IRI activity arises not because of insufficient binding, but rather because it becomes overgrown by the ice. The authors look to relate sustained IRI activity to the volume occupied by a polymer at the ice interface, and suggest that both enthalpic and entropic contributions may play a part in PVA's activity. Finally, they also examine the role small chain block copolymers, specifically polyvinylalcohol/polyvinylamine, could have in designing small, IRI-active polymers.

Ice nucleation and growth is an important research area with many areas of application, where the activity of polymers such as PVA and antifreeze proteins has drawn considerable interest from a broad range of experimental and theoretical studies. The work in this study generally appears to be well executed and the paper is generally well-presented. While I found this to be overall an interesting study making an important contribution to the field, I also identified some issues both with the work and with the presentation. However, on balance I believe that the study does provide sufficient new insights to warrant publication in Nature Communications. I recommend revision of the manuscript, where my more detailed and specific comments now follow.

1) Their definition of the orientation of the polymers appears to be used inconsistently in the manuscript and is very confusing. For example, the statement in the Fig. 1 caption on page 4, "the orientation with respect to the growing ice front" is not consistent with the images provided in Fig. 1(a); rather, assuming Fig. 1(a) is correct, the caption statement should be "the orientation with respect to the direction (z) of ice growth". Then in Fig. 3 on page 7, the images are not consistent with those in Fig. 1(a), for example Fig. 3(a) is described as being "parallel". Additionally, the behaviour in Fig. 6 on page 13, appears inconsistent with the orientations shown in Fig. 3. (In my attempt to understand the manuscript, I have assumed the orientations given in Fig. 1(a) are correct.) The authors must revise their manuscript to ensure the orientations are consistently and clearly defined throughout.

2) a) I found the discussion of enthalpic and entropic contributions to PVA-ice binding starting on page 14 somewhat confusing and apparently somewhat arbitrary. At the end of the first paragraph on page 14, the authors state "We therefore suggest additional driving forces may contribute to PVA-ice binding, other than the enthalpic gain from hydrogen bonding." Several lines later (in the next paragraph) they conclude "we therefore suggest that the PVA-ice interaction might thus benefit from an additional entropic gain." I am satisfied with the discussion to this point. It is the text that then follows, which includes a discussion of PVAm, that becomes problematic. The problems arise because of the simplistic approach taken here in comparing PVA and PVAm. While it is true that they differ only by the replacement of the hydroxyl groups by primary protonated amines, one might expect that the hydration structure of the two polymers could be considerably altered (e.g. due to the net positive charges on the PVAm, and the asymmetric H-bonding ability of the amines – only donors not acceptors – relative to OH functionality). Moreover their statement "As primary protonated amines can form hydrogen bonds - albeit not as readily as hydroxyl groups" is contradicted on page 18 by the statement "the fact that the primary protonated amines ... characterising the PVAm segment ... have higher binding energies associated with their hydrogen bonds". Hence the concluding statement on page 14 "that the predominant driving force of the PVA-ice interaction must be enthalpic in nature" does not obviously follow from the preceding text. Either the second half of this paragraph needs to be clarified and expanded, or this discussion simply dropped.

b) In view of the above, the discussion in the first paragraph on page 18, should include mention of the net positive charges on the amines and their asymmetric H-bond ability as being much less compatible to ice structure (relative to OH groups).

c) I did not find that the statement of page 21 "the interplay between enthalpy and entropy is probably skewed toward the former in the specific case of PVA" was supported by evidence. I would recommend it be dropped (unless clear evidence can be provided).

3) In Fig. 7, the authors report growth of an ice/water system with no polymer present. For this system the expected (average) linear growth rate is not observed. Moreover, their systems containing polymers that experience over-growth also show a similar slowing down of the growth rate. I was not convinced by the explanation for this behaviour provided at the bottom of page 12 and in the SI, namely that it is due to the finite extent of their water slabs. From Fig. 1(b) we can see that their systems are roughly 70 nm long, with a relatively narrow (~4 nm) initial slab of ice. This indicates that the initial slabs of water above the ice surface are more than 30 nm thick. Fig. 7(b) indicates that only about 10 nm of ice grow in their 200 ns trajectories, leaving 20 nm thick water slabs. Moreover, since the two systems compared in Fig. S8 differ only in their cross-sectional areas (but not their lengths), the overall slower growth in the smaller system would seem to be a finite-size effect in the x-y directions, not the z-direction. Thus statements (in SI) about the possible effects of the vacuum/liquid interface do not appear supported by the data. Rather, I suspect the effect is due to thermal gradients that develop in their very long systems. Due to their use of a homogeneous thermostat, their initial condition of an ice slab in the middle of their simulation box, and the rather large amount of heat liberated during ice growth, we can expect a temperature gradient to develop in their system, where the middle of the slab will become hot relative to the edges of the slab, and thus the temperature at the growing interface will not be equal to the average over the system. The presence of such a thermal gradient and the temperature at the growing interface would be reasonably straightforward to measure in their simulations. I would recommend that they re-examine the origins of this effect. I would also comment that the growth rates they initially observe, of about 1 Å/ns, are similar to values reported in previous simulation studies. While I would tend to agree with the authors that the lack of a linear growth regime is not likely to have qualitatively affected their results for polymer activities, it would nonetheless be helpful (for knowledgeable readers) to have discussed the origins for this issue more directly (I would recommend doing this in their SI).

4) On page 12, the authors discuss the possible influence of the effective volume on the activities of their polymers. Presumably this effective volume will be related to the area of the ice surface impacted by the polymer. Hence it might be illustrative to also show images of their systems (e.g. from Fig. 3) in cross-section, i.e. an x-y slice at the ice surface.

5) At several points in the manuscript (e.g. bottom of page 13), the difference of their results from those of previous workers is highlighted, although no explanation is provided. Without a suggestion, one is left to assume that one of the sets of simulations may be in error. Another possibility might be that the difference is due to model dependence (since the work in Refs. 22 and 24 uses the coarse-grain mW model for water.) The authors should provide some comment on the apparent difference in behaviour.

6) Several additional minor corrections.

a) On page 10, Fig. 8 is referenced before Figures 6 and 7.

b) Figures 3, 4, 7 and 9 should note that only water molecules in ice have been shown.

c) Ref. 57 is missing.

d) I noted a couple of issues in the SI, namely: section I.F should not appear the "experimental methods" section; there is a formatting issue at the top of page 5.

Reviewer #2 (Remarks to the Author):

This manuscript reports a molecular dynamics (MD) study that is augmented by experiments on the effects of PVAs on ice recrystallisation. Although the topic is interesting and the findings can have practical relevance in cryobiology, I didn't find the study convincing. In what follows I express my concerns, and hope the authors can address them.

First and foremost, the study aims to investigate the influence of PVAs on ice recrystallisation, which happens via ripening (i.e. small ice crystals become larger crystals). However, the MD simulations are based on ice growth from undercooled liquid water. One would think the two processes, ripening and solidification, follow distinct mechanisms. The authors should justify why their approach can shed light on the ice recrystallisation.

The authors show that PVA20 (with 20 monomers) is more potent than PVA10 when it comes to inhibiting ice growth. This is not surprising, considering a PVA20 molecule is roughly equivalent to 2 times PVA10. It will be much more meaningful if the authors base the comparison on simulations/experiments using the amount of PVAs with the same total number of monomers. This author did include such simulation (Fig S6) but the two PVA10 polymers were placed sequentially instead in parallel. I suspect that putting the two PVA10s in parallel will yield similar inhibition activity as a single PVA20.

I'm concerned about finite size effects. The authors only tested the effects on the pure water system. However, the study shows the PVAs inhibit ice growth by binding to the ice-water interface. If one enlarges the surface area of the interface, how would the results change?

Although the authors show that PVA slows down ice growth, the rate is still very fast even when PCA hasn't been overgrown: about 10 angstrom in 50 ns (left and mid panel of Fig 4), which is 20 cm per second. I doubt if such an ice growth rate is realistic in cryobiology.

I don't quite get the point of the first part of the result (Fig 2). This section seems to just suggest that PVA coils up in liquid water (at 300 K). This is, of course, not surprising because of the obvious conformational entropy gain for the polymer to not stay straight. Moreover, it is difficult to see why this has implications on the main topic of the paper, the influence of PVAs on ice recrystallisation.

Technical questions:

TiP4P-Ice has notoriously low diffusivity especially at undercooled conditions (e.g. <https://aip.scitation.org/doi/full/10.1063/1.4897524>). Would this affect the ice growth rate that the authors were measuring?

Did authors observe the formation of stacking faults (<https://www.pnas.org/content/112/34/10582>) during ice growth?

The authors selected one crystallographic direction of the ice front when simulating ice growth. Would other directions make a difference?

Would the authors consider using a stochastic model to model the ice growth rate in Fig.5 (see: <https://aip.scitation.org/doi/10.1063/1.4905955> notice that Eqn.13 is missing a minus sign).

I find Fig 10 rather confusing. What does each colored line represent?

Bingqing Cheng

Reviewer #3 (Remarks to the Author):

In this study, the authors use molecular simulations (and some experiments) to understand the ice recrystallisation inhibition (IRI) activity of poly(vinyl)alcohol (PVA) and related polymers. In particular, the authors use molecular simulations to investigate whether seeded ice growth can be impeded by PVA molecules (Fig. 1), which are restrained to different configurations and orientations, as well as by PVA molecules with different degrees of polymerization (DP).

Using the same simulation box dimensions in each case, the authors find that a PVA molecule with DP=20 arrests ice growth when it is restrained in its collapsed configuration; an extended configuration of PVA20 also arrests ice growth when it is constrained to be parallel to the ice-water interface, but when it is constrained to be perpendicular to the interface, PVA20 is engulfed by the growing ice front. The authors also find that PVA10 tends to get engulfed by ice. Finally, the authors

find that block copolymers of PVA and polyvinylamine (PVAm, which itself has insignificant IRI activity) can arrest ice growth with the PVA block binding to ice, and the PVAm block resisting engulfment. Based on their results, the authors argue that it is the “effective volume of PVA at the interface with ice” and not its lattice matching to ice, which dictates its IRI strength.

Although the manuscript makes use of creative molecular simulations to uncover interesting insights into the IRI activity of PVA, as detailed below, their interpretation of the results leaves much to be desired. In particular, the authors neglect the role of the distance between neighboring ice-binding molecules, which plays an important role in their ability to pin the ice-water interface and arrest ice growth; see ref. [1] below. Thus, I am unable to recommend publication of the manuscript in its present form.

Detailed Comments:

1) Although the results obtained in Figs. 3-5 are illuminating, they did not constitute the full picture when it comes to IRI activity. In particular, PVA molecules with different configurations and/or degree of polymerization would bind to ice with different affinities, and could therefore be present at the ice-water interface at different concentrations; differences in surface concentration would lead to different average distances between PVA molecules, which would influence IRI [1].

2) Fig. 5 highlights that in some simulations, PVA10 is overgrown by ice and in others, it arrests ice growth. What is special, if anything, about the simulations that arrest ice growth? Does PVA-ice binding occur in a certain orientation in those simulations?

3) The authors argue that ice growth is unaffected by doubling the concentration of PVA10, and refer to Figure S6 for further details. However, the simulation setup used in Fig. S6 is not quite realistic. An increased bulk concentration of PVA would also result in greater binding to ice and reduce the net distance between adsorbed PVA molecules, which would then impact IRI activity.

4) The authors argue that “the weaker IRI activity of short polymers such as PVA10 compared to PVA20 is due to the shorter polymer becoming overgrown by the advancing ice front, rather than - as previously thought - the limited number of hydroxyl groups binding to ice.” While the former is certainly a factor in lowering the IRI activity of PVA, as shown here, the latter could also be an important determinant of IRI activity through its influence on binding affinity and surface concentration [1].

5) The authors state that: “We show that PVA is a very flexible polymer which can bind to ice in any conformation. This is a very important result, as it removes a long-standing design constraint, that of lattice matching, in the synthesis of IRI active polymers.” This statement is too strong: lattice matching may not be necessary, but what is required for a polymer to bind ice remains non-trivial. In other words, knowing that PVA binds to ice (in spite of not being lattice matched) is insufficient to infer which other polymers would bind ice.

6) The results shown in Fig. 6 do not provide sufficiently strong support for the notion that IRI activity depends solely on the “effective volume of PVA at the interface with ice volume”. For one, Fig. 5 does not show a one-to-one relationship between the number of ice molecules at end of a simulation and the volume occupied by the polymer; moreover, the former is not a great measure of IRI activity.

7) The authors assert that: “In investigating the driving forces responsible for the PVA-ice interaction, we have found that, in addition to the enthalpy gain originating from the hydrogen bonds between PVA and ice, entropic contributions due to the de-solvation of methylene groups can also play a role.” Although the arguments that the authors make are sensible, they do not present the appropriate results to support their arguments. What the authors show (in Fig. 7) is that when PVA binds to ice, its methylene groups lose some of their liquid-like hydration waters. However, the replacement of liquid-like water molecules in the PVA hydration shell by ice-like waters is simply a natural consequence of the binding process, and does not bear upon entropic/enthalpic contributions to the free energetics of binding.

8) The authors state that: “As primary protonated amines can form hydrogen bonds - albeit not as

readily as hydroxyl groups - we suggest that the predominant driving force of the PVA-ice interaction must be enthalpic in nature."

Once again, while reasonable, this statement is not supported by the authors' results.

Minor Comments:

1) In the simulation setup described in Fig. 1 (and for Fig. 3b in particular), it would be useful to clarify whether it is possible for PVA to be excluded from ice. For example, was PVA position restrained to be at a particular distance away from initial position of the ice-water interface? Was there was enough space in the simulation box for PVA to move away from the ice front?

2) Results in Fig. 10 might be easier to appreciate if recast in the form of probabilities.

3) Typo on page 21, line 7: PVA20

Reference:

[1] P. M. Naullage, Y. Qiu, and V. Molinero, "What Controls the Limit of Supercooling and Superheating of Pinned Ice Surfaces?", *J. Phys. Chem. Lett.* 9, 1712 (2018).

Note: references in **green** can be found at the end of this document.

- Response to Reviewer #1 –

This paper reports molecular dynamic (MD) simulations aimed at investigating the ice recrystallisation inhibition (IRI) activity of poly(vinyl)alcohol (PVA). Specifically, using atomistic models for water and PVA in their simulations, the authors systematically probe the impact of conformation, number of functional groups, and degree of polymerization. In contrast to previous studies, they find that PVA can bind ice in any conformation and that lattice matching is not a prerequisite for ice inhibition. They also show that short-chain PVA's reduced IRI activity arises not because of insufficient binding, but rather because it becomes overgrown by the ice. The authors look to relate sustained IRI activity to the volume occupied by a polymer at the ice interface and suggest that both enthalpic and entropic contributions may play a part in PVA's activity. Finally, they also examine the role small chain block copolymers, specifically polyvinyl alcohol / polyvinylamine, could have in designing small, IRI-active polymers. Ice nucleation and growth is an important research area with many areas of application, where the activity of polymers such as PVA and antifreeze proteins has drawn considerable interest from a broad range of experimental and theoretical studies. The work in this study generally appears to be well executed and the paper is generally well-presented. While I found this to be overall an interesting study making an important contribution to the field, I also identified some issues both with the work and with the presentation. However, on balance I believe that the study does provide sufficient new insights to warrant publication in Nature Communications. I recommend revision of the manuscript, where my more detailed and specific comments now follow.

Our reply: We thank the Reviewer for this insightful, positive response.

Reviewer's comment: 1) Their definition of the orientation of the polymers appears to be used inconsistently in the manuscript and is very confusing. For example, the statement in the Fig. 1 caption on page 4, "the orientation with respect to the growing ice front" is not consistent with the images provided in Fig. 1(a); rather, assuming Fig. 1(a) is correct, the caption statement should be "the orientation with respect to the direction (z) of ice growth". Then in Fig. 3 on page 7, the images are not consistent with those in Fig. 1(a), for example Fig. 3(a) is described as being "parallel". Additionally, the behaviour in Fig. 6 on page 13, appears inconsistent with the orientations shown in Fig. 3. (In my attempt to understand the manuscript, I have assumed the orientations given in Fig. 1(a) are correct.) The authors must revise their manuscript to ensure the orientations are consistently and clearly defined throughout.

Our reply: We do apologise for having mislabelled the caption in Fig. 3, which in turn meant that Fig. 1(a) and Fig. 6 became unclear. In order to fix this issue and provide a consistent definition of the orientation of the polymers, we have:

- Modified the caption of Fig. 1 as: "[...] for the linear polymers, their orientation with respect to the direction (z) of the growing ice front [...]"
- Modified the caption of Fig. 3 as follows: "a) An extended PVA chain ($R_g \sim 1$) is restrained so as to be perpendicular with respect to the growing ice front. b) A similarly extended conformation is restrained so as to be parallel with respect to the growing ice front. c) A compact, globular conformation ($R_g \sim 0.5$)."

Reviewer's comment: 2) a) I found the discussion of enthalpic and entropic contributions to PVA-ice binding starting on page 14 somewhat confusing and apparently somewhat arbitrary. At the end of the first paragraph on page 14, the authors state "We therefore suggest additional driving forces may contribute to PVA-ice binding, other than the enthalpic gain from hydrogen bonding." Several lines later (in the next paragraph) they conclude "we therefore suggest that the PVA-ice interaction might thus benefit from an additional entropic gain." I am satisfied with the discussion to this point. It is the text that then follows, which includes a discussion of PVAm, that becomes problematic. The problems arise because of the simplistic approach taken here in comparing PVA and PVAm. While it is true that they differ only by the replacement of the hydroxyl

groups by primary protonated amines, one might expect that the hydration structure of the two polymers could be considerably altered (e.g. due to the net positive charges on the PVAm, and the asymmetric H-bonding ability of the amines – only donors not acceptors – relative to OH functionality). Moreover, their statement “As primary protonated amines can form hydrogen bonds - albeit not as readily as hydroxyl groups” is contradicted on page 18 by the statement “the fact that the primary protonated amines ... characterising the PVAm segment ... have higher binding energies associated with their hydrogen bonds”. Hence the concluding statement on page 14 “that the predominant driving force of the PVA-ice interaction must be enthalpic in nature” does not obviously follow from the preceding text.

Either the second half of this paragraph needs to be clarified and expanded, or this discussion simply dropped.

b) In view of the above, the discussion in the first paragraph on page 18, should include mention of the net positive charges on the amines and their asymmetric H-bond ability as being much less compatible to ice structure (relative to OH groups).

c) I did not find that the statement of page 21 “the interplay between enthalpy and entropy is probably skewed toward the former in the specific case of PVA” was supported by evidence. I would recommend it be dropped (unless clear evidence can be provided).

Our reply: We thank the Reviewer for these comments, which prompted us to perform additional simulations and strengthen our discussion re: the role of entropic contributions.

a) Our original statement that “the primary protonated amines [...] characterising the PVAm segment [...] have higher binding energies associated with their hydrogen bonds” is incorrect, in that by “higher” we meant “more positive”. We apologise for the confusion. Indeed, the hydrogen bonding potential of primary protonated amines is inferior to that of hydroxyl groups. We have clarified this aspect by modifying the main text as follows [page 19]: **“The non-negligible IRI activity of PVA₅-b-PVAm₅ compared to PVA₁₀ originates from to the fact that the primary protonated amines such as the NH⁺₃ groups characterising the PVAm segment of PVA₅-b-PVAm₅ form fewer and weaker hydrogen bonds compared to hydroxyl groups (as discussed above in the case of PVAm).**

b) We agree – we have modified the discussion as illustrated below.

c) We agree that this claim lacked sufficient evidence. On the other hand, we believe this is an important point that deserves to be included in the manuscript. Hence, we have performed a new set of simulations of the IRI activity of polypropylene (PP), a purely hydrophobic polymer which – despite being insoluble in water in reality – allowed us to strengthen our findings in terms of the role of entropic contributions. As illustrated in Fig. S9 (reported below), we find that PP does indeed interact with the growing ice front via the desolvation of the methylene groups – the same mechanism we proposed to be relevant in the case of PVA. Interestingly, PP slows down ice growth to some extent, but it is much less effective than PVA, as can be seen in Fig. S9. This is consistent with our hypothesis that entropic contributions originating from the desolvation of – in this case – methylene groups do play a role in the PVA-ice interaction, albeit they are far less important than the enthalpic contributions due to hydrogen bonding. These findings are consistent with the work of *Naullage et al.* [1] which suggests that “a small entropic drive towards binding of ice which may come from hydrophobic desolvation of the methylene groups”. Finally, we are currently in the process of publishing a work on the IRI activity of amphiphilic peptides where the balance between enthalpic and entropic contributions plays a crucial role; thus, we believe that this discussion is of great relevance to different classes of IRI-active compounds. To address these points we have added Fig. S9 in the Supplementary Information (SI) and we have modified the discussion in the main text as follows [page 15]: **“At this stage, it is interesting to consider the possibility that the PVA-ice interaction might benefit from a small additional entropic gain. This hypothesis is motivated by the ever-growing evidence suggesting that hydrophobic interactions play a pivotal role in the context of IRI activity (see e.g. Refs. 31–34). Thus, we have investigated the desolvation of the methylene (-CH₂) hydrophobic groups of PVA, which might lead to an entropic contribution in addition to the enthalpic gain originating from the PVA-ice hydrogen bonding. In solution, the first solvation shell of the methylene groups contains on average five water molecules (see Fig. S1 in the SI as well as the Methods section). The de-solvation of methylene groups upon their interaction with ice is illustrated**

in Fig. 7d and corresponds to the drop in the number of water molecules within the solvation shell and the corresponding increase of ice molecules. As the solvation shell of methylene is structurally constrained compared to the bulk liquid phase, there is an entropic contribution associated with the desolvation of the $-\text{CH}_2$ groups.

Figure S9. Comparison of the IRI activity of PVA₂₀ and polypropylene (PP). The latter is far less efficient than the former in slowing down ice growth (lower panel), as it interacts with ice by means of purely hydrophobic interactions (see inset, where we look at desolvation of the methylene groups). Interestingly, the desolvation of the methylene groups of PP is similar to that we observe in the case of PVA (upper panel). The green lines refer to the ice growth observed in absence of any polymer. Note that the upper panel and inset are reproduced in the main text as Fig. 5 and Fig. 7.

The latter can form hydrogen bonds but not as readily as hydroxyl groups. This is due to the asymmetric nature of the hydrogen bonding in protonated amines: that is, they are more likely to offer donors, than acceptors [31]. Furthermore, the PVAm hydration shell structure is likely to be affected by net positive charge of the amine group and hence if we take these factors into consideration it serves to explain why we observe only minimal interaction with the growing ice front for PVAm compared to PVA. In turn this indicates that the predominant driving force for the binding of PVA to ice is mostly enthalpic in nature, i.e. comes from its hydroxyl groups.

Overall, we are in no position to quantify this subtle balance between enthalpy and entropy. However, in light of all of the above we can confidently put forward that entropic contributions can play a role in the context of IRI activity. This is important not just in the context of IRI-active polymers but to e.g., amphiphilic ice-binding proteins and other systems as well. As an example, we have very recently investigated the IRI activity of synthetic peptides characterized by a strong interplay between enthalpic and entropic contributions [2].

To probe this possibility in greater detail, we have investigated the IRI activity of polypropylene (PP), an entirely hydrophobic polymer. While PP is in reality insoluble in water, this computational experiment allows us to probe the potential effect of the desolvation of methylene groups in absence of hydrogen bonding. We find that PP does indeed interact with the growing ice front via purely hydrophobic interactions originating from the desolvation of methylene groups (see Fig. S9). Unlike PVA, PP doesn't stop the growth of the ice front, but it can slow it down somewhat. This mechanism is identical to that reported for PVA in Fig. 7d and it is consistent with the work of Naullage *et al.* [21] which also suggests the existence of a small entropic contribution due to the hydrophobic desolvation of PVA methylene groups.

In the attempt to unravel the interplay between enthalpic and entropic contributions, we have also investigated the IRI activity of polyvinylamine (PVAm, DP = 20), via MD simulations as well as splat-assay measurements, summarized in Fig. 8. We find the IRI activity of PVAm to be substantially lower than that of PVA. The main difference between PVA and PVAm is that in PVAm the hydroxyl groups are replaced by primary protonated amines.

Reviewer's comment: 3) In Fig. 7, the authors report growth of an ice/water system with no polymer present. For this system the expected (average) linear growth rate is not observed. Moreover, their systems containing polymers that experience over-growth also show a similar slowing down of the growth rate. I was not convinced by the explanation for this behaviour provided at the bottom of page 12 and in the SI, namely that it is due to the finite extent of their water slabs. From Fig. 1(b) we can see that their systems are roughly 70 nm long, with a relatively narrow (~4 nm) initial slab of ice. This indicates that the initial slabs of water above the ice surface are more 30 nm thick. Fig. 7(b) indicates that only about 10 nm of ice grow in their 200 ns trajectories, leaving 20 nm thick water slabs. Moreover, since the two systems compared in Fig. S8 differ only in their cross-sectional areas (but not their lengths), the overall slower growth in the smaller system would seem to be a finite-size effect in the x-y directions, not the z-direction. Thus statements (in SI) about the possible effects of the vacuum/liquid interface do not appear supported by the data. Rather, I suspect the effect is due to thermal gradients that develop in their very long systems. Due to their use of a homogeneous thermostat, their initial condition of an ice slab in the middle of their simulation box, and the rather large amount of heat liberated during ice growth, we can expect a temperature gradient to develop in their system, where the middle of the slab will become hot relative to the edges of the slab, and thus the temperature at the growing interface will not be equal to the average over the system. The presence of such a thermal gradient and the temperature at the growing interface would be reasonably straightforward to measure in their simulations. I would recommend that they re-examine the origins of this effect. I would also comment that the growth rates they initially observe, of about 1 Å/ns, are similar to values reported in previous simulation studies. While I would tend to agree with the authors that the lack of a linear growth regime is not likely to have qualitatively affected their results for polymer activities, it would nonetheless be helpful (for knowledgeable readers) to have discussed the origins for this issue more directly (I would recommend doing this in their SI).

Our reply: We apologise for the confusion. The entire system is ~70 nm in the z dimension, which includes vacuum in the +/- z directions (see the Methods section, page 25). The water slabs are in fact 11 nm thick (not 30 nm) in both the +/- z directions and the ice grows up to 8-10 nm (meaning that we are left with roughly 1-3 nm of liquid phase). In addition, regarding Fig. S8, there is no difference re: the cross-sectional areas of the systems; the difference is instead in terms of the length (i.e. the extent along the z-axis) of the simulation boxes. That is, the top panel has a z dimension of ~11 nm whereas the bottom panel has z dimension of ~6.5 nm. We apologise that this is very unclear in the SI (part vi) and have rephrased this section to also include a clearer figure (Fig.S8 below) as well as the following text [page 5 SI]:

Figure S8. Progress of the ice front as a function of time for systems containing water slabs of different thickness. The extent of the linear ice growth regime depends on the thickness of the water slab.

“Fig.S8 depicts the growth curve for three different simulation boxes (note that the xy dimensions are kept constant but the water slab increases from 6.5 nm to 11 nm and 22 nm respectively in the z-direction). We see that a water slab of 6.5 nm thickness has virtually no linear growth due to the slowing dynamics of the water molecules at the vacuum/liquid interface and is due to the finite size effect for a system this size. Increasing to 11 nm shows a comparatively better growth rate, which can be improved upon further, if we add a 22 nm water slab to the system. In this case we observe a completely linear growth regime. Due to the available computational resources, studying the IRI activity of PVA in a simulation box with a completely linear growth regime is regrettably

unfeasible (~167,000 atoms), and furthermore does not change the underlying observations made in this paper. As such we had to strike a balance and opted for the system with a 11nm water slab.”

We agree with the Reviewer that we should have verified that the deviation of the growth rate from the expected linear regimes is indeed due to the limited extent of the liquid phase. To address this issue, we have investigated the ice growth rate as a function of the initial thickness of the water slab. The results are reported in Fig. S8 and unequivocally show that indeed thicker water slabs result in a perfectly linear regime. Crucially, as pointed out by the Reviewer themselves, the deviation from linearity does not have an impact on our results. We have verified this explicitly by repeating some of our simulations (for PVA₁₀ and PVA₂₀ as well) using thicker water slabs – we have observed no changes with respect to our original results (this can be seen in Fig.S10 below and in the SI) .

Reviewer’s comment: 4) On page 12, the authors discuss the possible influence the effective volume on the activities of their polymers. Presumably this effective volume will be related to the area of the ice surface impacted by the polymer. Hence it might be illustrative to also show images if their systems (e.g. from Fig. 3) in cross-section, i.e. an x-y slice at the ice surface.

Our reply: Following the Reviewer’s suggestion, we have estimated the extent of the area of the ice surface impacted by the polymers. To this end, we have selected sections of the molecular dynamics trajectories corresponding to situations where the polymer is interacting with the ice surface. We have then computed the area of the alpha shape (i.e., a generalization of the concept of convex hull for a non-necessarily-convex set) constructed from the 2D set of x and y coordinates of the PVA atomic positions (see methodologies). This is a rather rough estimate (as it does not take into account *explicitly* the roughness of the ice surface) that nonetheless proved to be quite informative, as illustrated in the revised version of Fig. 6 (reported below). In fact, we have chosen to include this analysis in the main text alongside the information about the volume of the polymer.

Fig. 6. The IRI activity of PVA depends on the effective volume of the polymer and its contact area with the ice surface. a) IRI activity of PVA, quantified as the number of ice molecules found at the end of our simulations (re-scaled so that the initial state of the system corresponds to the position in time $t-t_0$ when the polymer binds to ice), as a function of the average volume occupied by the polymer. Empty circles and crosses refer to a situation where the polymer has become overgrown (o/g) by the ice front or it does show substantial IRI activity, respectively. b) Average ice-PVA contact area for different PVA conformations c) Representative cross-sections (xy-plane) of the system upon binding of PVA.

To take into account this new analysis we have modified the main text as follows [page 13]: “**We can further elaborate on the impact of the different binding orientations by considering the contact area between**

ice and PVA, which we have estimated according to the approach detailed in the Methods section. For instance, combining the information about volume (Fig. 6a) and contact area (Fig. 6b) we can understand the difference in terms of IRI activity between the “parallel” and “perpendicular” orientations of the linear PVA conformation (see Fig. 6c). While both orientations are characterized by a very similar occupied volume, the parallel orientation spans a much smaller contact area compared to the perpendicular one. As a result, the parallel orientation is much more likely to get overgrown by the ice front, which results in a negligible IRI activity.

Reviewer’s comment: 5) *At several points in the manuscript (e.g. bottom of page 13), the difference of their results from those of previous workers is highlighted, although no explanation is provided. Without a suggestion, one is left to assume that one of the sets of simulations may be in error. Another possibility might be that the difference is due to model dependence (since the work in Refs. 22 and 24 uses the coarse-grain mW model for water.) The authors should provide some comment on the apparent difference in behaviour.*

Our reply: Previous work utilising the mW model [3] has indeed suggested a 2:1 binding pattern for PVA on ice, which we do not observe in our simulations. This contradiction is likely due to the fact that the investigation reported in Ref. [1]: (a.) assumed that PVA behaves like a linear molecule (it does not); (b.) used of a coarse-grained model of water (which does not capture the complexity of the hydrogen bond network) (c.) simulated the growing ice front as a static, smooth crystalline surface (i.e., no actual ice growth). We have summarised these differences in the main text as follows [Page 6]: **“Previous computational work has indeed identified a 2:1 binding pattern for PVA on ice, which however we do not observe in our simulations. This contradiction is likely due to the fact that the investigation reported in Ref. [22]: (a.) assumed that PVA behaves like a linear molecule when bidding to ice, which appears not to be the case at the supercooling investigated here; (b.) utilized the mW [30] and a united-atom forcefield [31] to model water molecules and PVA, respectively. The coarse-grained nature of these force fields is unlikely to capture the complexity of hydrogen bonding at heterogeneous interfaces such as the PVA-ice one. We used an fully atomistic forcefield instead, as discussed in the Methods section; (c.) the distance between the two binding hydroxyls characterizing this 2:1 pattern was thought to be comparable to the nearest oxygen-oxygen distance characterizing the *pristine* crystalline structure of the primary prismatic ice surface. As no simulations of the actual growth of the ice front were included (i.e. PVA bound on a static ice surface only) the roughness of the growing ice front is not taken into account. In contrast, in this work we have considered the PVA interactions with a dynamically growing ice front.”**

- Response to Reviewer #2 -

This manuscript reports a molecular dynamics (MD) study that is augmented by experiments on the effects of PVAs on ice recrystallisation. Although the topic is interesting and the findings can have practical relevance in cryobiology, I didn't find the study convincing. In what follows I express my concerns and hope the authors can address them.

Our reply: We have thoroughly addressed the concerns of the Reviewer. We do sincerely hope that our efforts, particularly in terms of the extensive sets of additional simulations we have performed to strengthen our claims, will serve to convince the Reviewer of the soundness of our findings.

Reviewer's comment: *First and foremost, the study aims to investigate the influence of PVAs on ice recrystallisation, which happens via ripening (i.e. small ice crystals becomes larger crystals). However, the MD simulations are based on ice growth from undercooled liquid water. One would think the two processes, ripening and solidification, follow distinct mechanisms. The authors should justify why their approach can shed light on the ice recrystallisation.*

Our reply: The experimental reality of the ice re-crystallization process is that there always is a fraction of un-frozen, supercooled liquid water in between the ice crystals. In fact, this is the very reason why experimental investigations of the IRI activity usually involve the addition of NaCl in the samples – so as to ensure the existence of an “eutectic” where ice and supercooled liquid water co-exist [4]. As such, performing molecular dynamics simulations of ice growth in contact with supercooled liquid water is entirely justified as one of the best approaches to probe the IRI ability of additives such as, in this case, PVA. Of course, our simulations provide an *indirect* assessment of the IRI activity of PVA in that we cannot simulate the coalescence of different ice grains in the presence of PVA. However, the microscopic process ultimately responsible for the slow down re: the Ostwald ripening can be traced down to the PVA-ice interactions which are the subject of our study. To further support this claim, we note that our results are in excellent agreement with the experimental reality investigated in this work in terms of splat assay, thus confirming the suitability of our methodology to address IRI activity. To clarify this point, we have added the following sentence into the revised version of the manuscript [page 26]: **“We note that the ice re-crystallization process involves the coalescence of different ice crystals embedded in supercooled liquid water. Thus, it is appropriate to address the interaction between PVA and ice at the interface between the latter and supercooled liquid water. While our simulations provide an indirect assessment of the IRI activity of PVA (in that we are not simulating the coalescence of different ice crystals), the interaction between PVA and ice is ultimately the molecular-level mechanism responsible for the extent of ice re-crystallization. In fact, our simulations are in excellent agreement with the experimental splat assays (which utilize NaCl to ensure an equilibrium between the ice crystals and the supercooled liquid phase) discussed in the remainder of the paper.”**

Reviewer's comment: *The authors show that PVA20 (with 20 monomers) is more potent than PVA10 when it comes to inhibiting ice growth. This is not surprising, considering a PVA20 molecule is roughly equivalent to 2 times PVA10. It will be much more meaningful if the authors base the comparison on simulations/experiments using the amount of PVAs with the same total number of monomers. The authors did include such simulation (Fig S6) but the two PVA10 polymers were placed sequentially instead in parallel. I suspect that putting the two PVA10s in parallel will yield similar inhibition activity as a single PVA20.*

Our reply: We do apologise for the confusion. The original version of Fig. S6a shows the starting configuration of the system *before equilibration*. That was the wrong figure to include – we have now revised Fig. S6a to clarify that, indeed, the two molecules of PVA₁₀ are not restrained and are able to diffuse independently throughout the water slab. As such, having two PVA₁₀ polymers in the same system does not yield a similar inhibition activity to that of a single PVA₂₀ (as shown in the second panel of Fig. S6 below [Fig. S6b in the SI]). This is because the polymers diffuse independently and therefore also bind to ice at independent times. Consequently, PVA₁₀ still gets overgrown. However statistically unlikely, though, we might encounter

Figure S6. Upper panel: computational setup of the system containing two PVA₁₀ polymers in each water slab. Lower panel: representative snapshot of a simulation highlighting the two PVA₁₀ polymers diffusing and binding (as well as becoming eventually overgrown by the growing ice front) in an independent fashion.

a situation where two PVA₁₀ polymers bind *at exactly the same* time to the growing ice surface. While we haven't explored this possibility directly, we do now know that a single PVA₁₀ interacting with an ice surface of $\sim 26 \text{ nm}^2$ shows the very same IRI activity of a single PVA₁₀ interacting with an ice surface of $\sim 13 \text{ nm}^2$. This latter scenario effectively corresponds to two PVA₁₀ polymers interacting with an ice surface of $\sim 26 \text{ nm}^2$ at the same time and thus serves to prove the point that indeed, two PVA₁₀ polymers are less effective in terms of IRI activity if compared to a single PVA₂₀ (see Fig.S10 below). We have clarified this point by adding the following sentence in the revised version of the manuscript [page 11]: **“This conclusion holds even if we consider the statistically unlikely scenario by which the two PVA₁₀ polymers bind to the ice surface at exactly the same point in time. While we haven't explored this possibility directly, we do know that a single PVA₁₀ interacting with an ice surface of $\sim 26 \text{ nm}^2$ shows the very same IRI activity of a single PVA₁₀ interacting with an ice surface of $\sim 13 \text{ nm}^2$ (based on simulations carried out with smaller box boundary dimension and therefore smaller surface area, see Fig.S10). This latter scenario effectively corresponds to two PVA₁₀ polymers interacting with an ice surface of $\sim 26 \text{ nm}^2$ at the same time and thus serves to prove the point that indeed, two PVA₁₀ polymers are less effective in terms of IRI activity if compared to a single PVA₂₀.”**

Reviewer's comment: *I'm concerned about finite size effects. The authors only tested the effects on the pure water system. However, the study shows the PVAs inhibit ice growth by binding to the ice-water interface. If one enlarges the surface area of the interface, how would the results change?*

Our reply: We thank the Reviewer for highlighting this important aspect. It is very true that the IRI activity of any given compound strongly depends on the surface “coverage”, i.e. how many (in this case) polymers bind to the growing ice front per unit area. In the original version of our manuscript, we limited our analysis to a specific surface area (size M in Fig. S10 below). Following the Reviewer's comment, we have now investigated the IRI activity of both PVA₁₀ and PVA₂₀ when interacting with ice surface of different extent: 13.3 nm^2 (size S), 26.1 nm^2 (size M), 53.4 nm^2 (size M/L) and 124.1 nm^2 (size L). The results are summarised in Fig. S10 below. As expected, for very large values of surface area (i.e. size L) the effect of PVA₂₀ is minimal: after all, even the most potent IRI agents are inactive under a certain surface coverage. Experimentally, this evidence translates into the observation that the activity of IRI agents depends on their concentration in solution (see for example Fig.8 in main text). According to our simulations, PVA₂₀ is IRI active for surface coverages of ~ 0.04 polymers / nm^2 or higher: this is an important piece of information that is not easily accessible

experimentally. In fact, the IRI activity of both PVA₂₀ and PVA₁₀ increases in the case of size S albeit PVA₂₀ remain much more effective than PVA₁₀ even in this regime.

Figure S10. The IRI activity of PVA₁₀ and PVA₂₀ as a function of surface area: 13.3 nm² (size S), 26.1 nm² (size M), 53.4 nm² (size M/L) and 124.1 nm² (size L). The blue lines correspond to trajectories in which the polymer gets overgrown, while grey lines highlight situations where substantial IRI activity is observed. Green lines correspond to ice growth in the absence of any polymer.

To clarify this issue, we have added the following discussion in the revised version of the manuscript [page 13]: “Finally, we also note that the activity of any given IRI agent strongly depends on its coverage of the ice surface: in this case, lower numbers of PVA polymers per (ice) unit area would lead to a lower IRI activity. This is the reason why, experimentally, a certain IRI agent displays activity above a certain concentration but it is not straightforward to translate (bulk) IRI agent concentration to the effective ice surface coverage. To explore this aspect, we have investigated the IRI activity of both PVA₁₀ and PVA₂₀ as a function of different surface areas, namely 13.3 nm² (size S), 26.1 nm² (size M, the “main” size which we refer to throughout the paper), 53.4 nm² (size M/L) and 124.1 nm² (size L). The results are summarized in Fig.S10 and suggest that PVA₂₀ is IRI active for surface coverages of ~ 0.04 polymers / nm² or higher. In addition, the IRI activity of both PVA₂₀ and PVA₁₀ increases in the case of size S (compared to size M), albeit PVA₂₀ remain much more effective than PVA₁₀ even in this regime.

Reviewer's comment: *I don't quite get the point of the first part of the result (Fig 2). This section seems to just suggest that PVA coils up in liquid water (at 300 K). This is, of course, not surprising because of the obvious conformational entropy gain for the polymer to not stay straight. Moreover, it is difficult to see why this has implications on the main topic of the paper, the influence of PVAs on ice recrystallisation.*

Our reply: We agree with the Reviewer in that the fact that PVA tends to be found in a random coil conformation is to be expected. And yet, this simple piece of information has been up to now ignored in the context of the IRI activity of small polymers. As an example, in the recent work of Naullage *et al.* [3] PVA is assumed to be found chiefly in a rather linear extended conformation. Crucially, the notion of flexible polymers as objects that straighten themselves upon binding to ice has somehow taken hold within the experimental community. In this work, we have invested some extra effort in conclusively proving that PVA has to be considered as a random coil when interacting with ice. This deceptively trivial detail is extremely important to support one of the main claims of this work, namely that the polymer can bind and show potential IRI activity in any conformation (not just linear) and further that the number of potential hydrogen bonds in principle available for e.g., PVA to interact with ice is less important than the occupied volume of the polymer on the ice front. This is a radically different concept with respect to the previous and present literature and hinges on the knowledge that PVA is a random coil in solution. While we do agree with the Reviewer that this detail *per se* is not especially surprising, it is the cornerstone of a new way of thinking about polymer-ice interactions which brakes tradition with the venerable lattice match argument - according to which polymers bind to ice because of geometric reasons which in reality failed to materialize because of the flexibility of the polymers. We feel all of the above is explained rather clearly in the original version of the paper.

Reviewer's comment: *Although the authors show that PVA slows down ice growth, the rate is still very fast even when PCA hasn't been overgrown about 10 angstrom in 50 ns (left and mid panel of Fig 4), which is 20 cm per second. I doubt if such an ice growth rate is realistic in cryobiology [...] TiP4P-Ice has notoriously low diffusivity especially at undercooled conditions (e.g. <https://aip.scitation.org/doi/full/10.1063/1.4897524>). Would this affect the ice growth rate that the authors were measuring?*

Our reply: We agree with the Reviewer in that the self-diffusion coefficient of TIP4P/Ice is, at the mild supercooling ($T_m - T = 8$ K) investigated in this work, about one order of magnitude smaller than the experimental value. Having said that, the ice growth rate computed from our simulations is indeed ~ 0.7 Å / ns. This value is not only in excellent agreement with previous computational work employing the very same forcefield [5] but it is also comparable to the experimental growth rate (~ 0.2 m/s) at the same supercooling [6] While the growth rate predicted by TIP4P/Ice is indeed faster than the experimental one, we note that the choice of TIP4P/Ice over TIP4P/2005 (which diffusion coefficient is closer to the experimental value) is motivated by the fact that TIP4P/Ice provides a much more accurate description of water and ice polymorphs and a much more accurate melting point compared to TIP4P/2005. Indeed, TIP4P/Ice has been consistently used to study ice nucleation and growth and – importantly – we are confident that the CHARMM36 - TIP4P/Ice combination of force fields is well suited to perform MD simulations of biomolecules in supercooled water and in contact with ice, including the study of IRI active biomolecules (see refs 32,42,43,45-47 in main text). To clarify this aspect, we have added the following sentence in the revised version of the manuscript [page 6 of the SI]: **“We note that, while the self-diffusion coefficient of TIP4P/Ice water is lower than the experimental value measured at the supercooling of 8 K investigated in this work, the ice growth rate we have computed from our simulations (~ 0.7 Å/ns) is comparable to the experimental value of ~ 0.2 Å/ns reported in Ref. [1]. While the growth rate predicted by TIP4P/Ice is indeed faster than the experimental one, we note that the choice of the TIP4P/Ice over e.g. TIP4P/2005 water model (which diffusion coefficient is closer to the experimental value) is motivated by the fact that (a.) TIP4P/Ice provides a much more accurate description of water and ice polymorphs and a much more accurate melting point compared to TIP4P/2005. Indeed, TIP4P/Ice has been consistently used to study ice nucleation and growth in the past [see e.g. Ref [2]] (b.) we are confident that the CHARMM36 - TIP4P/Ice combination of force fields is well suited to perform MD simulations of biomolecules in**

supercooled water and in contact with ice, including the study of IRI active biomolecules (see Refs. 32,42,43,45-47 found in main text)”.

Reviewer’s comment: *Did authors observe the formation of stacking faults (<https://www.pnas.org/content/112/34/10582>) during ice growth?*

Our reply: No, we did not observe the emergence of staking faults in our simulations. We have chosen not to comment on this aspect in the manuscript as we do not believe it might have an impact on our results.

Reviewer’s comment: *The authors selected one crystallographic direction of the ice front when simulating ice growth. Would other directions make a difference?*

Our reply: We have chosen to focus on a specific crystallographic direction because of the experimental work by Budke and Koop [Chem. Phys. Chem, 7, 2601 (2006)] which shows that PVA interacts with the primary prismatic face of hexagonal ice. In fact, the primary prismatic face was also the main focus of the work on PVA by Naullage *et al.* [J. Phys. Chem 48, 26949 (2017)] as well as the Wang *et al.* [ACS Lang. 34, 5116 (2018)]. To clarify this aspect, we have modified the main text as follows [page 25]: **“The experimental evidence reported by Budke and Koop [26] demonstrates that PVA interacts with the primary prismatic face of hexagonal ice. Thus, in this work we have chosen to focus on this particular surface, in line with previous computational work [22,23]”.**

Reviewer’s comment: *Would the authors consider using a stochastic model to model the ice growth rate in Fig.5 (see:<https://aip.scitation.org/doi/10.1063/1.4905955> notice that Eqn.13 is missing a minus sign).*

Our reply: This is an intriguing suggestion. However, as discussed in detail above, the growth rate we have obtained from our simulations is in excellent agreement with previous computational work as well as the experimental data. As such, we do not believe that embarking into an alternative approach to compute the ice growth rate would be beneficial to this work.

Reviewer’s comment: *I find Fig 10 rather confusing. What does each colored line represent?*

Our reply: Apologies for the confusion, each line merely represents an independent trajectory. We have re-done the plots in a monochromatic colour scheme and have additionally re-cast the figures as PDFs as per Reviewer #3 comment.

- Response to Reviewer #3 -

In this study, the authors use molecular simulations (and some experiments) to understand the ice recrystallisation inhibition (IRI) activity of poly(vinyl)alcohol (PVA) and related polymers. In particular, the authors use molecular simulations to investigate whether seeded ice growth can be impeded by PVA molecules (Fig. 1), which are restrained to different configurations and orientations, as well as by PVA molecules with different degrees of polymerization (DP). Using the same simulation box dimensions in each case, the authors find that a PVA molecule with DP=20 arrests ice growth when it is restrained in its collapsed configuration; an extended configuration of PVA₂₀ also arrests ice growth when it is constrained to be parallel to the ice-water interface, but when it is constrained to be perpendicular to the interface, PVA₂₀ is engulfed by the growing ice front. The authors also find that PVA₁₀ tends to get engulfed by ice. Finally, the authors find that block copolymers of PVA and polyvinylamine (PVAm, which itself has insignificant IRI activity) can arrest ice growth with the PVA block binding to ice, and the PVAm block resisting engulfment. Based on their results, the authors argue that it is the “effective volume of PVA at the interface with ice” and not its lattice matching to ice, which dictates its IRI strength. Although the manuscript makes use of creative molecular simulations to uncover interesting insights into the IRI activity of PVA, as detailed below, their interpretation of the results leaves much to be desired. In particular, the authors neglect the role of the distance between neighboring ice-binding molecules, which plays an important role in their ability to pin the ice-water interface and arrest ice growth; see ref. [1] below. Thus, I am unable to recommend publication of the manuscript in its present form.

Our reply: We appreciate the concerns of the Reviewer. We have thoroughly addressed every issue that has been raised, which special emphasis on the role of the distance between neighbouring molecules. To that end, we have performed several new sets of simulations which enabled us to strengthen our claims. We hope that the Reviewer will acknowledge our efforts and recognise the soundness of our results.

Reviewer’s comment: *1) Although the results obtained in Figs. 3-5 are illuminating, they did not constitute the full picture when it comes to IRI activity. In particular, PVA molecules with different configurations and/or degree of polymerization would bind to ice with different affinities and could therefore be present at the ice-water interface at different concentrations; differences in surface concentration would lead to different average distances between PVA molecules, which would influence IRI [1].*

Our reply: We agree with the Reviewer in that surface concentration (and thus the average distance between PVA molecules at the ice surface) is a key aspect, which we have investigated further in this revised version of the manuscript - see our answer to comment 3) below. In terms of binding affinities, though, our findings show that PVA binds to ice quite effectively notwithstanding its configuration or its degree of polymerization. Coiled and linear conformations both bind to ice – it is the difference in terms of the area/volume they occupy at the PVA-ice interface which seems to be regulating the extent of their IRI activity. The same holds for different degrees of polymerization: PVA₁₀ and PVA₂₀ both bind to ice, but – given the same surface concentration (more on that below) – the area/volume they occupy at the interface with the growing ice front is very different, which leads to very different IRI activities for the two polymers. As a specific example, having two PVA₁₀ polymers in the same system (i.e. double the surface concentration in terms of molecules / ice unit area) does not yield a similar IRI activity to that of a single PVA₂₀ (as shown in the second panel of Fig. S6 above [Fig. S6b in the SI]). This is true when the two PVA₁₀ polymers are free to bind independently to ice and even when considering the statistically unlikely scenario where the two PVA₁₀ polymers would bind to ice at exactly the same time. While we have not explored that possibility directly, we know that a single PVA₁₀ interacting with an ice surface of $\sim 26 \text{ nm}^2$ shows the very same IRI activity of a single PVA₁₀ interacting with an ice surface of $\sim 13 \text{ nm}^2$. This latter scenario effectively corresponds to two PVA₁₀ polymers interacting with an ice surface of $\sim 26 \text{ nm}^2$ at the same time and thus serves to prove the point that indeed, two PVA₁₀ polymers are less effective in terms of IRI activity if compared to a single PVA₂₀ (see Fig.S10). We have clarified this point by adding the following sentence in the revised version of the manuscript [page 11]: **“This conclusion holds even if we consider the statistically unlikely scenario by which the two PVA₁₀ polymers bind to the ice surface at exactly the same point in time. While we haven’t explored this possibility directly, we do know that a single PVA₁₀ interacting with an ice surface of $\sim 26 \text{ nm}^2$ shows the very same**

IRI activity of a single PVA₁₀ interacting with an ice surface of ~ 13 nm² (based on simulations carried out with smaller box boundary dimension and therefore smaller surface area, see Fig.S10). This latter scenario effectively corresponds to two PVA₁₀ polymers interacting with an ice surface of ~ 26 nm² at the same time and thus serves to prove the point that indeed, two PVA₁₀ polymers are less effective in terms of IRI activity if compared to a single PVA₂₀.”

Reviewer’s comment: 2) *Fig. 5 highlights that in some simulations, PVA10 is overgrown by ice and in others, it arrests ice growth. What is special, if anything, about the simulations that arrest ice growth? Does PVA-ice binding occur in a certain orientation in those simulations.*

Our reply: This is a really good point - we thank the Reviewer for bringing this to our attention. Upon further investigation, we found a correlation between the IRI activity of PVA₁₀ and the minimum distance (rMI hereafter) between the periodic images of the polymer. Specifically, PVA₁₀ polymers (in the size M box, see Fig.S10 above) which became overgrown by the ice front are characterised, on average, by $rMI = 3.7 \pm 0.30$, while PVA₁₀ polymers which showed some extent of IRI activity are characterised by an average value of $rMI = 3.5 \pm 0.30$ nm. This trend holds for different surface areas as well: in particular, we have obtained, in the case of the size S box (see Fig. S10 above), $rMI = 2.6 \pm 0.16$ nm and $rMI = 2.3 \pm 0.22$ nm for inactive and active PVA10 polymers, respectively. These trends are indicative of the role of the (polymer) surface coverage with respect to the (ice) surface, an aspect we have thoroughly addressed in this revised version of the manuscript (see below). To account for this finding, we have modified the last paragraph on page 11 to read as follows (added text in **bold** page 11): **“Thus, we argue that the IRI activity of PVA10 is largely affected by the kinetics of ice growth i.e., the time period during which PVA10 manages to hold back the ice front before becoming engulfed in the ice phase might result in some IRI activity observed experimentally - but due to its eventual engulfment in ice its activity will be much weaker compared to PVA20. Interestingly, we have found a correlation between the IRI activity of PVA10 and the minimum distance (rMI hereafter) between the periodic images of the polymer. Specifically, PVA10 polymers which became overgrown by the ice front are characterised, on average, by $rMI = 3.7 \pm 0.30$, while PVA10 polymers which showed some extent of IRI activity are characterised by an average value of 3.5 ± 0.30 nm . This trend holds for different surface areas as well: in particular, we have obtained, in the case of the size S box ((see Fig. S10, label S for further details)), $rMI = 2.6 \pm 0.16$ nm and $rMI = 2.3 \pm 0.22$ nm for inactive and active PVA10 polymers, respectively. These trends are indicative of the key role of the (polymer) surface coverage with respect to the (ice) surface – an aspect we discuss below.**

Reviewer’s comment: 3) *The authors argue that ice growth is unaffected by doubling the concentration of PVA10 and refer to Figure S6 for further details. However, the simulation setup used in Fig. S6 is not quite realistic. An increased bulk concentration of PVA would also result in greater binding to ice and reduce the net distance between adsorbed PVA molecules, which would then impact IRI activity.*

Our reply: We do apologise for the confusion. The original version of Fig. S6a shows the starting configuration of the system *before equilibration*. That was the wrong figure to include – we have now revised Fig. S6a to clarify that, indeed, the two molecules of PVA₁₀ are not restrained and are able to diffuse independently throughout the water slab. It is very true that an increased bulk concentration of PVA₁₀ would result in a higher surface concentration of the polymer on the ice surface. However, we have found that having two PVA₁₀ polymers in the same system does not yield a similar inhibition activity to that of a single PVA₂₀ (as shown in the second panel of Fig. S6 below [Fig. S6b in the SI]). As discussed in our answer to comment 2) above, this holds even when assuming that the two PVA₁₀ would bind at the same time, which is in any case statistically unlikely. In order to investigate the impact of the distance between neighbouring ice-binding molecules, an important point raised by the Reviewer which is directly related to the issue of surface concentration, we have performed several new sets of molecular dynamics simulations. The lateral dimensions of our simulation boxes in the original version of the paper was 4.5 x 5.8 nm (a surface concentration S_c of 0.038 molecules / nm²). We have now investigated the IRI activity of both PVA₁₀ and PVA₂₀ using boxes with later dimensions of 13.3 nm² (size S), 26.1 nm² (size M), 53.4 nm² (size M/L) and 124.1 nm² (size L). The

results are illustrated in Fig. S10 on page 9 of this Rebuttal. As expected, for very large values of surface area (i.e. size L) the effect of PVA₂₀ is minimal: after all, even the most potent IRI agents are inactive under a certain surface coverage. Experimentally, this evidence translates into the observation that the activity of IRI agents depends on their concentration in solution (see Fig.8 in the main paper as an example) According to our simulations, PVA₂₀ is IRI active for surface coverages of ~ 0.04 polymers / nm² or higher: this is an important piece of information that is not easily accessible experimentally. In fact, the IRI activity of both PVA₂₀ and PVA₁₀ increases in the case of size S albeit PVA₂₀ remain much more effective than PVA₁₀ even in this regime. An aspect that we did not consider but we know it is important (see e.g. ACS Macro Lett. 2019, 8, 1063-1067) and it could be addressed in future studies is the effect of aggregation. To clarify this issue, we have added the following discussion in the revised version of the manuscript [page 13]: **“Finally, we also note that the activity of any given IRI agent strongly depends on its coverage of the ice surface: in this case, lower numbers of PVA polymers per (ice) unit area would lead to a lower IRI activity. This is the reason why, experimentally, a certain IRI agent displays activity above a certain concentration but it is not straightforward to translate (bulk) IRI agent concentration to the effective ice surface coverage. To explore this aspect, we have investigated the IRI activity of both PVA₁₀ and PVA₂₀ as a function of different surface areas, namely 13.3 nm² (size S), 26.1 nm² (size M, the “main” size which we refer to throughout the paper), 53.4 nm² (size M/L) and 124.1 nm² (size L). The results are summarized in Fig. S10 and suggest that PVA₂₀ is IRI active for surface coverages of ~ 0.04 polymers / nm² or higher. In addition, the IRI activity of both PVA₂₀ and PVA₁₀ increases in the case of size S (compared to size M), albeit PVA₂₀ remain much more effective than PVA₁₀ even in this regime.**

Reviewer’s comment: 4) *The authors argue that “the weaker IRI activity of short polymers such as PVA10 compared to PVA20 is due to the shorter polymer becoming overgrown by the advancing ice front, rather than - as previously thought - the limited number of hydroxyl groups binding to ice.” While the former is certainly a factor in lowering the IRI activity of PVA, as shown here, the latter could also be an important determinant of IRI activity through its influence on binding affinity and surface concentration [J. Phys. Chem. Lett. 9, 1712 (2018)].*

Our reply: We agree with the Reviewer. While we have found that PVA seems to bind “irreversibly” (consistently with the results of Naullage *et al.* [1]) no matter its conformation or its degree of polymerization at the specific supercooling we are working at, the binding affinity might become relevant in cases where the kinetics of ice formation is such that PVA can bind and unbind within a relevant time scale. To acknowledge this possibility, we have highlighted the following sentence in the revised version of the manuscript [page 6]: **“While this holds at this particular supercooling only (8 K), [...] We note, however, that the binding affinity of the polymer might become relevant in conditions where the kinetics of ice formation is slow enough for PVA to bind/unbind reversibly.”**

Reviewer’s comment: 5) *The authors state that: “We show that PVA is a very flexible polymer which can bind to ice in any conformation. This is a very important result, as it removes a long-standing design constraint, that of lattice matching, in the synthesis of IRI active polymers.” This statement is too strong: lattice matching may not be necessary, but what is required for a polymer to bind ice remains non-trivial. In other words, knowing that PVA binds to ice (in spite of not being lattice matched) is insufficient to infer which other polymers would bind ice.*

Our reply: This statement does not advocate the absence of lattice matching as a feature common to all IRI active polymers. We agree with the Reviewer that lattice matching can play a rather important role in some cases – we wanted to clarify that you do not *need* lattice matching to have an IRI active polymer. In other words, we do not want to propose a constraint in terms of the design of IRI agents, on the contrary we want to remove a specific constraint. We also know (from ongoing work) that lattice matching becomes more and more relevant when dealing with (a.) large and (b.) rigid IRI agents. Thus, we have modified this sentence as follows [page 23]: **“[...] It is important to stress that lattice matching might be playing an important role for different classes of polymers and/or different types of IRI agents. In particular, the recent body of**

work on ice-binding proteins [38-40] suggest that lattice matching is key when dealing with large and rigid IRI agents, as opposed to small, flexible polymers like PVA₁₀₋₂₀.”

Reviewer’s comment: 6) The results shown in Fig. 6 do not provide sufficiently strong support for the notion that IRI activity depends solely on the “effective volume of PVA at the interface with ice volume”. For one, Fig. 5 does not show a one-to-one relationship between the number of ice molecules at end of a simulation and the volume occupied by the polymer; moreover, the former is not a great measure of IRI activity.

Our reply: We agree with the referee in that the number of ice molecules at the end of a simulation is a sub-optimal measure of IRI activity. Please note, however, that we have re-scaled that number so as to keep into account the onset of the PVA-ice interaction. That is, the fact that we are scaling the time axis (see e.g. Fig. 5 in the main text) so that the zero corresponds to the time when PVA binds to ice should mitigate the qualitative nature of this observable. On the other hand, the difference in the ice growth rate between cases where PVA manages to stop the progress of the ice front (grey lines in e.g. Fig. 5) and cases where PVA gets overgrown by the crystal (blue lines in e.g. Fig.5) is quite stark. We also agree with Reviewer that our statement re: the role of the volume of the polymer is too strong. However, we have now estimated the extent of the area of the ice surface impacted by the polymers as well. To this end, we have selected sections of the molecular dynamics trajectories corresponding to situations where the polymer is interacting with the ice surface. We have then computed the area of the alpha shape (i.e., a generalization of the concept of convex hull for a non-necessarily-convex set) constructed from the 2D set of x and y coordinates of the PVA atomic positions. This is a rather rough estimate (as it does not take into account *explicitly* the roughness of the ice surface) that nonetheless proved to be quite informative, as illustrated in the revised version of Fig. 6 (reported above). In fact, we have chosen to include this analysis in the main text alongside the information about the volume of the polymer, so as to strengthen our claims. Nevertheless, we have followed the Reviewer’s suggestion and toned down our original statement as follows [page 17]: **“In light of our findings, which show that the low IRI activity of PVA₁₀ is largely dictated by its small effective volume / contact area at the ice-water interface [...]”**.

Reviewer’s comment: 7) The authors assert that: “In investigating the driving forces responsible for the PVA-ice interaction, we have found that, in addition to the enthalpy gain originating from the hydrogen bonds between PVA and ice, entropic contributions due to the de-solvation of methylene groups can also play a role.” Although the arguments that the authors make are sensible, they do not present the appropriate results to support their arguments. What the authors show (in Fig. 7) is that when PVA binds to ice, its methylene groups lose some of their liquid-like hydration waters. However, the replacement of liquid-like water molecules in the PVA hydration shell by ice-like waters is simply a natural consequence of the binding process and does not bear upon entropic/enthalpic contributions to the free energetics of binding.

Our reply: We agree that this claim lacked sufficient evidence. Hence, we have performed a new set of simulations of the IRI activity of polypropylene (PP), a purely hydrophobic polymer which – despite being insoluble in water in reality – allowed us to strengthen our findings in terms of the role of entropic contributions. As illustrated in Fig. S9 (reported above), we find that PP does indeed interact with the growing ice front via the desolvation of the methylene groups – the same mechanism we proposed to be relevant in the case of PVA. Interestingly, PP slows down ice growth to some extent, but it is much less effective than PVA, as can be seen in Fig. S9. This is consistent with our hypothesis that entropic contributions originating from the desolvation of – in this case – methylene groups do play a role in the PVA-ice interaction, albeit they are far less important than the enthalpic contributions due to hydrogen bonding. These findings are consistent with the work of Naullage *et al.* [J. Phys. Chem 48, 26949 (2017)] which suggests that “a small entropic drive towards binding of ice which may come from hydrophobic desolvation of the methylene groups”. Finally, we are currently in the process of publishing a work on the IRI activity of amphiphilic peptides where the balance between enthalpic and entropic contributions plays a crucial role; thus, we believe that this discussion is of great relevance to different classes of IRI-active compounds. Finally, while it is true that the desolvation of the -CH₂ groups can very well be a natural consequence of the PVA binding (via hydrogen bonding) that does not mean that said desolvation might not contribute to the free energy of binding. To address these points we have

added Fig. S9 in the Supplementary Information (SI) and we have modified the discussion in the main text as follows [page 15]: “At this stage, it is interesting to consider the possibility that the PVA-ice interaction might benefit from a small additional entropic gain. This hypothesis is motivated by the ever-growing evidence suggesting that hydrophobic interactions play a pivotal role in the context of IRI activity (see e.g. Refs. 31–34). Thus, we have investigated the desolvation of the methylene (-CH₂) hydrophobic groups of PVA, which might lead to an entropic contribution in addition to the enthalpic gain originating from the PVA-ice hydrogen bonding. In solution, the first solvation shell of the methylene groups contains on average five water molecules (see Fig. S1 in the SI as well as the Methods section). The desolvation of methylene groups upon their interaction with ice is illustrated in Fig. 7d and corresponds to the drop in the number of water molecules within the solvation shell and the corresponding increase of ice molecules. As the solvation shell of methylene is structurally constrained compared to the bulk liquid phase, there is an entropic contribution associated with the desolvation of the -CH₂ groups.

To probe this possibility in greater detail, we have investigated the IRI activity of polypropylene (PP), an entirely hydrophobic polymer. While PP is in reality insoluble in water, this computational experiment allows us to probe the potential effect of the desolvation of methylene groups in absence of hydrogen bonding. We find that PP does indeed interact with the growing ice front via purely hydrophobic interactions originating from the desolvation of methylene groups (see Fig. S9). This mechanism is identical to that reported for PVA in Fig. 7d and it is consistent with the work of Naullage *et al.* [22] which also suggests the existence of a small entropic contribution due to the hydrophobic desolvation of PVA methylene groups.

In the attempt to unravel the interplay between enthalpic and entropic contributions, we have also investigated the IRI activity of polyvinylamine (PVAm, DP = 20), via MD simulations as well as splat-assay measurements, summarized in Fig. 8. We find the IRI activity of PVAm to be substantially lower than that of PVA. The main difference between PVA and PVAm is that in PVAm the hydroxyl groups are replaced by primary protonated amines. The latter can form hydrogen bonds but not as readily as hydroxyl groups. This is due to the asymmetric nature of the hydrogen bonding in protonated amines: that is, they are more likely to offer donors, than acceptors [31]. Furthermore, the PVAm hydration shell structure is likely to be affected by net positive charge of the amine group and hence if we take these factors into consideration it serves to explain why we observe only minimal interaction with the growing ice front for PVAm compared to PVA. In turn this indicates that the predominant driving force for the binding of PVA to ice is mostly enthalpic in nature, i.e. comes from its hydroxyl groups.

Overall, we are in no position to quantify this subtle balance between enthalpy and entropy. However, in light of all of the above we can confidently put forward that entropic contributions can play a role in the context of IRI activity. This is important not just in the context of IRI-active polymers but to e.g., amphiphilic ice-binding proteins and other systems as well. As an example, we have very recently investigated the IRI activity of synthetic peptides characterized by a strong interplay between enthalpic and entropic contributions [1].

Reviewer’s comment: 8) The authors state that: “As primary protonated amines can form hydrogen bonds - albeit not as readily as hydroxyl groups - we suggest that the predominant driving force of the PVA-ice interaction must be enthalpic in nature.” Once again, while reasonable, this statement is not supported by the authors’ results.

Our reply: We agree. We hope that our response to the previous Reviewer’s comment will serve to address this issue. In addition, we note that our original statement that “the primary protonated amines [...] characterizing the PVAm segment [...] have higher binding energies associated with their hydrogen bonds” is incorrect, in that by “higher” we meant “more positive”. We apologise for the confusion. Indeed, the hydrogen bonding potential of primary protonated amines is inferior to that of hydroxyl groups. We have clarified this aspect by modifying the main text as follows [page 19]: “The non- negligible IRI activity of PVA₅-b-PVAm₅ compared to PVA₁₀ originates from to the fact that the primary protonated amines

such as the NH_3^+ groups characterising the PVAm segment of PVA₅-b-PVAm₅ form fewer and weaker hydrogen bonds compared to hydroxyl groups (as discussed above in the case of PVAm).

Reviewer's comment: 1) In the simulation setup described in Fig. 1 (and for Fig. 3b in particular), it would be useful to clarify whether it is possible for PVA to be excluded from ice. For example, was PVA position restrained to be at a particular distance away from initial position of the ice-water interface? Was there was enough space in the simulation box for PVA to move away from the ice front?

Our reply: The position of PVA was not restrained to be at a particular distance away from the initial position of the ice water interface. In fact, for the production runs we have collected 20 different trajectories with different PVA configurations at different starting positions with respect to the initial ice-water interface. The PVA could, in principle, move away from the ice: in fact, in the case of polymers with very little IRI activity (e.g. the PVAm we have investigated in this work) we have seen cases where the polymer has been “pushed” by the growing ice front up to – almost - the water-vacuum interface. “Almost”, as we did set a constraint for the centre of mass of the polymer to be at least 1 nm below the water-vacuum interface. Apologies for not making this clear, we have included this information on page 26 as follows: “[...]the polymers were not restrained so during equilibration they were free to diffuse the entire water slab [...] The starting configurations of PVA in the water slab depended on the last frame in the equilibration run and as such all trajectories had independent starting configurations. To prevent the polymers interacting with the water/vacuum boundary, a constraint for the centre of mass of the polymer was set to be at least 1 nm below the water-vacuum boundary layer.”

Reviewer's comment: Results in Fig. 10 might be easier to appreciate if recast in the form of probabilities.

Our reply: Thank you for your suggestion. The figure has now been recast as a PDF and the colours have been substituted for a monochromatic colour scheme.

References (within this Rebuttal):

- [1] P. M. Naullage, L. Lupi, and V. Molinero, *J. Phys. Chem C*, **121**, 26949 (2017).
- [2] C.A. Stevens, F. Bachtiger, X. Kong, L.A. Abriata, G.C. Sosso, M.I. Gibson and H. Klok. “A Minimalistic Cyclic Ice-Binding Peptide from Phage Display”. *Nature Communications* – under revision.
- [3] V. Molinero and E. B. Moore, *J. Phys. Chem. B*, **113**, 4008 (2009).
- [4] C. A. Knight, D. Wen, R. A. Laursen, *Cryobiology*, **32**, 23 (1995).
- [5] P. Montero de Hijes, J. R. Espinosa, C. Vega, and E. Sanz, *J. Chem. Phys.*, **151**, 044509 (2019).
- [6] Y. Xu, N. G. Petrik, R. S. Smith, B. D. Kay, and G. A. Kimmel, *PNAS*, **113**, 14921 (2016).

REVIEWERS' COMMENTS

Reviewer #1 (Remarks to the Author):

Having read the authors' response letter and reviewed the revised manuscript and supporting material, and I find that the revisions the authors performed, which include results from additional simulations, have reasonably addressed all of my previous comments and concerns, and has resulted in a significantly improved paper. Hence, I now support the publication of this manuscript.

I did note one minor point to bring to the authors' attention: in Fig. S9 the authors have not defined the green lines (I assume they are the result for systems with no PVA).

Reviewer #2 (Remarks to the Author):

The authors have substantially improved their analysis and presentation of the result. I think the manuscript is now suitable for publication.

Bingqing Cheng

Reviewer #3 (Remarks to the Author):

The authors have addressed some of points previously raised by this reviewer (detailed comments # 2, 3, 5 and all minor points). However, they have not satisfactorily addressed the bulk of critique, which I lump into three groups below.

Points 1 and 4: Dependence of IRI activity on binding affinity for ice

The authors have performed additional calculations with different simulation box sizes to show that molecules are overgrown by ice more easily when larger simulation boxes are used. However, this fact stems directly from Gibbs-Thomson equation, and was never in question. In fact, it is precisely this insight, which calls into question whether the authors' results bear directly upon IRI activity.

The authors appear to have missed the point regarding the impact that binding affinity can have on IRI activity. e.g., under the assumption that surface concentration is proportional to the adsorption constant, and that the bulk concentrations of PVA20 and PVA10 are identical, the ratio of the surface concentrations of PVA10 and PVA20 would be $\exp(dG)$, where dG is the difference in the binding free energies of PVA20 and PVA10.

If we further assume that each PVA monomer contributes 1 kT to ice binding, then $dG = 10$ kT, and the ratio of surface concentrations would be roughly 22,000. The key point here is that in contrast with the surface concentration range explored by the authors (which spans 1 order of magnitude), small changes in binding affinity can drastically alter surface concentrations, and thereby have a dramatic influence on IRI activity.

That said, I agree that appropriately accounting for differences in binding affinity is a non-trivial task. Moreover, it may very well be that under certain circumstances, the ice-water interface becomes saturated with PVA molecules.

However, in lieu of making ambiguous statements, such as: "In terms of binding affinities, though, our findings show that PVA binds to ice quite effectively notwithstanding its configuration or its degree of polymerization."; I would urge the authors to be clear on their position.

Is it the authors' contention that the different molecules studied here are expected to have the same surface coverage? If so, they should state such an assertion clearly, and provide evidence to support it. If not, the authors' should make it clear that they are able to investigate only one

(important) aspect of IRI activity (pertaining to propensity for engulfment), but that their results do not bear directly upon experimental measurements.

Point 6: Volume and/or area at interface as a molecular measure of IRI activity

The notion that a solute that occupies low volume/surface area at the ice-water interface will have poor IRI activity or conversely that a solute that occupies high volume/surface area at the ice-water interface will have better IRI activity is exceedingly simplistic. For example, such a criterion can't be used to design new IRIs. Most reasonably long polymers will occupy high volume/surface area at the ice-water interface, and most of them will also have poor IRI activity.

Points 7 and 8: Enthalpic vs Entropic contributions to ice binding

In response to the critique in point 7, the authors perform ice-binding calculations on polypropylene, which is a purely hydrophobic polymer. However, in doing so, and in their subsequent arguments, they entirely miss the point that when a solute binds to ice, whether it is PVA or PP, it has to exchange some of its hydration waters for ice. The fact that it does so has no bearing on whether binding is enthalpic or entropic in nature.

In the revised text, the authors state: "Overall, we are in no position to quantify this subtle balance between enthalpy and entropy. However, in light of all of the above we can confidently put forward that entropic contributions can play a role in the context of IRI activity."

These somewhat conflicting statements and the surrounding discussion do not contribute to this manuscript. Either the authors should quantify the relative enthalpic and entropic contributions to ice binding or remove any statements, confident or otherwise, on such contributions.

- Response to Reviewer #3 –

Reviewer's comment: *Points 1 and 4: Dependence of IRI activity on binding affinity for ice.* The authors have performed additional calculations with different simulation box sizes to show that molecules are overgrown by ice more easily when larger simulation boxes are used. However, this fact stems directly from Gibbs-Thomson equation, and was never in question. In fact, it is precisely this insight, which calls into question whether the authors' results bear directly upon IRI activity. The authors appear to have missed the point regarding the impact that binding affinity can have on IRI activity. e.g., under the assumption that surface concentration is proportional to the adsorption constant, and that the bulk concentrations of PVA₂₀ and PVA₁₀ are identical, the ratio of the surface concentrations of PVA₁₀ and PVA₂₀ would be $\exp(dG)$, where dG is the difference in the binding free energies of PVA₂₀ and PVA₁₀. If we further assume that each PVA monomer contributes 1 kT to ice binding, then $dG = 10$ kT, and the ratio of surface concentrations would be roughly 22,000. The key point here is that in contrast with the surface concentration range explored by the authors (which spans 1 order of magnitude), small changes in binding affinity can drastically alter surface concentrations, and thereby have a dramatic influence on IRI activity. That said, I agree that appropriately accounting for differences in binding affinity is a non-trivial task. Moreover, it may very well be that under certain circumstances, the ice-water interface becomes saturated with PVA molecules. However, in lieu of making ambiguous statements, such as: "In terms of binding affinities, though, our findings show that PVA binds to ice quite effectively notwithstanding its configuration or its degree of polymerization."; I would urge the authors to be clear on their position. Is it the authors' contention that the different molecules studied here are expected to have the same surface coverage? If so, they should state such an assertion clearly, and provide evidence to support it. If not, the authors should make it clear that they are able to investigate only one (important) aspect of IRI activity (pertaining to propensity for engulfment), but that their results do not bear directly upon experimental measurements.

Our reply: The substantial difference hypothesised by Reviewer #3 in terms of the binding affinity between PVA₁₀ and PVA₂₀ relies on the assumption that *all* the PVA monomers would bind to ice. This is not the case: our results demonstrated instead that only a fraction of the PVA monomers is involved in the interaction with the ice front. For instance, for PVA₂₀ only ~6-8 monomers out of 20 bind, on average, to ice, which means that the binding affinities of PVA₁₀ and PVA₂₀ are indeed very similar. In fact, given that PVA tends to be found in a coiled, compact conformation (as we have showed in this work), the binding affinity of this polymer does *not* scale linearly with its chain length – as suggested by Reviewer #3. To clarify this point we have added the following sentences to the revised version of the manuscript (page 10): **"In light of these results, it is important to clarify the role of the "binding affinity" of PVA with respect to ice, which is often thought of as a linear function of the number of monomers [24, 26]. As PVA tends to be found in the form of a random coil, the number of the monomers interacting with the ice surface does not increase linearly with the polymer length. Indeed, the binding affinities of PVA₁₀ and PVA₂₀, albeit difficult to quantify when dealing with dynamically growing ice surfaces, appears to be very similar."**

Reviewer's comment: *1) Point 6: Volume and/or area at interface as a molecular measure of IRI activity.* The notion that a solute that occupies low volume/surface area at the ice-water interface will have poor IRI activity or conversely that a solute that occupies high volume/surface area at the ice-water interface will have better IRI activity is exceedingly simplistic. For example, such a criterion can't be used to design new IRIs. Most reasonably long polymers will occupy high volume/surface area at the ice-water interface, and most of them will also have poor IRI activity.

Our reply: We agree with the Reviewer in that size alone is not a sensible criterion to infer the IRI activity of any given compound. However, provided that the compound in question shows a strong interaction with the ice phase (such in the case of PVA), then the contact area between the IRI compound and ice becomes important and, as we have showed in this work, correlates quite clearly with the IRI activity. To clarify this point, we have added the following sentence to the revised version of the manuscript (page 13): **"Thus, it is**

important to clarify that neither molecular weight nor the extent of the average contact area between any given IRI agent and the ice phase can be used, in isolation, to infer the IRI activity of polymers and anti-freeze proteins alike. Instead, it is the interplay between the volume and/or surface occupied by PVA and the ability of the latter to interact with the ice surface that dictates its overall efficiency as an IRI agent.”

Reviewer’s comment: **Points 7 and 8: Enthalpic vs Entropic contributions to ice binding.** *In response to the critique in point 7, the authors perform ice-binding calculations on polypropylene, which is a purely hydrophobic polymer. However, in doing so, and in their subsequent arguments, they entirely miss the point that when a solute binds to ice, whether it is PVA or PP, it has to exchange some of its hydration waters for ice. The fact that it does so has no bearing on whether binding is enthalpic or entropic in nature. In the revised text, the authors state: “Overall, we are in no position to quantify this subtle balance between enthalpy and entropy. However, in light of all of the above we can confidently put forward that entropic contributions can play a role in the context of IRI activity.” These somewhat conflicting statements and the surrounding discussion do not contribute to this manuscript. Either the authors should quantify the relative enthalpic and entropic contributions to ice binding or remove any statements, confident or otherwise, on such contributions.*

Our reply: PP is a purely hydrophobic polymer. In those cases where we have observed an interaction between PP and ice (signified by a reduced ice growth rate), this interaction must, by definition, be driven by entropy. This is because PP does not provide any opportunity for enthalpic interactions with ice (or indeed anything else) nor does it feature particularly strong/localised charges that might lead to electrostatic interactions. Our statement: “Overall, we are in no position to quantify this subtle balance between enthalpy and entropy. However, in light of all of the above we can confidently put forward that entropic contributions can play a role in the context of IRI activity.” is not contradictory: we cannot put a number on the entropic gain associated with the PVA -CH₂ interaction with ice but given our investigation of PP we can safely say that entropy does play a role. Thus, we believe this argument to be sufficiently clear already in the present version of the manuscript.